# Multifidelity Simulation-based Inference for Computationally Expensive Simulators

**Anastasia N. Krouglova**[1,2,3], **Hayden R. Johnson**[1,2,3], **Basile Confavreux**[4],
**Michael Deistler**[5,6,7], **Pedro J. Gonçalves**[1,2,3,8]

[1] Department of Computer Science, KU Leuven, Belgium
[2] VIB Center for AI and Computational Biology (VIB.AI), Leuven, Belgium
[3] VIB-KU Leuven Center for Neuroscience, Belgium
[4] Gatsby Computational Neuroscience Unit, UCL, London, UK
[5] Machine Learning in Science, University of Tübingen, Tübingen, Germany
[6] Tübingen AI Center, Tübingen, Germany
[7] Max Planck Institute for Biological Intelligence, Martinsried, Germany
[8] Department of Electrical Engineering, KU Leuven, Belgium
`{nastya.krouglova, pedro.goncalves}@kuleuven.be`

## Abstract

Across many domains of science, stochastic models are an essential tool to understand the mechanisms underlying empirically observed data. Models can be of different levels of detail and accuracy, with models of high-fidelity (i.e., high accuracy) to the phenomena under study being often preferable. However, inferring parameters of high-fidelity models via simulation-based inference is challenging, especially when the simulator is computationally expensive. We introduce a multifidelity approach to neural posterior estimation that uses transfer learning to leverage inexpensive low-fidelity simulations to efficiently infer parameters of high-fidelity simulators. Our method applies the multifidelity scheme to both amortized and non-amortized neural posterior estimation. We further improve simulation efficiency by introducing a sequential variant that uses an acquisition function targeting the predictive uncertainty of the density estimator to adaptively select high-fidelity parameters. On established benchmark and neuroscience tasks, our approaches require up to two orders of magnitude fewer high-fidelity simulations than current methods, while showing comparable performance. Overall, our approaches open new opportunities to perform efficient Bayesian inference on computationally expensive simulators.

## 1 Introduction

Stochastic models are used across science and engineering to capture complex properties of real systems through simulations (Barbers et al., 2024; Nelson & Pei, 2021; Pillow & Scott, 2012; Marlier et al., 2021). These simulators encode domain-specific knowledge and provide a means to generate high-fidelity synthetic data, enabling accurate forward modeling of experimental outcomes. However, inferring model parameters from observed data can be challenging, especially when simulators are stochastic, the likelihoods of the simulators are inaccessible, or when simulations are computationally expensive.

Simulation-based inference (SBI) addresses these challenges by leveraging forward simulations to infer the posterior distribution, enabling quantification of uncertainty even when the likelihood is intractable (Cranmer et al., 2020). The challenge of extending sampling-based SBI methods like Approximate Bayesian Computation (ABC) (Tavaré et al., 1997; Pritchard et al., 1999) to problems with large numbers of parameters has driven significant advancements in neural-based approaches that estimate the likelihood (Papamakarios et al., 2019), the likelihood-to-evidence ratio (Hermans et al., 2020), or directly the posterior (Greenberg et al., 2019; Lueckmann et al., 2017; Papamakarios & Murray, 2016). In particular, amortized Neural Posterior Estimation (NPE) trains a neural density estimator to directly approximate the posterior, bypassing the need to estimate the model evidence

(Papamakarios & Murray, 2016). To improve inference for a fixed observation and allow stable training, truncated sequential variants have been introduced for neural posterior estimation (TSNPE) (Deistler et al., 2022), and neural ratio estimation (Miller et al., 2021). These approaches have leveraged recent progress in neural density estimation to improve the scalability and accuracy of SBI, allowing parameter inference in problems with higher dimensionality than was previously achievable (Ramesh et al., 2021; Gloeckler et al., 2024). Despite these advancements, SBI methods face computational challenges for scenarios involving expensive simulations or high-dimensional parameter spaces, as state-of-the-art methods often require extensive simulation budgets to achieve reliable posterior estimates (Lueckmann et al., 2021).

Multifidelity modeling offers a solution to this problem by balancing precision and efficiency. It combines accurate but costly high-fidelity models (Hoppe et al., 2021; Behrens & Dias, 2015) with faster, less accurate low-fidelity models. Here, low-fidelity models could be simplifications made possible through domain knowledge about the high-fidelity models, low-dimensional projection of the high-fidelity model, or surrogate modeling (Peherstorfer et al., 2018). For example, Reynolds-averaged Navier-Stokes (RANS) models simplify turbulent flow simulations in aerodynamics (Han et al., 2013), while climate models often reduce complexity by focusing on specific atmospheric effects (Held, 2005; Majda & Gershgorin, 2010). Similarly, mean-field approximations are used to capture certain features of spiking neural network dynamics (Vogels et al., 2011; Dayan & Abbott, 2001). Multifidelity methods have proven effective across domains—enhancing optimization through multifidelity Bayesian optimization (Song et al., 2019; Kandasamy et al., 2017), and improving the efficiency of inference through multifidelity Monte Carlo approaches (Peherstorfer et al., 2016; Nobile & Tesei, 2015; Giles, 2008; Zeng et al., 2023). In the context of SBI, we hypothesized that by leveraging the complementarity of high- and low-fidelity simulators, it would be possible to reduce the computational cost of inference while retaining inference accuracy.

In this work, we present MF-(TS)NPE, a multifidelity approach that improves the efficiency of amortized and non-amortized neural posterior estimation for expensive simulators. MF-(TS)NPE reduces the computational burden of posterior estimation by pre-training a neural density estimator on low-fidelity simulations and refining the inference with a smaller set of high-fidelity simulations. Additionally, we present MF-TSNPE-AF, an extension of MF-TSNPE with active learning, facilitating targeted parameter space exploration to effectively enhance high-fidelity posterior estimates given single observations. We focus on multifidelity cases where both models are simulators and where the low-fidelity model is a simplified version of the high-fidelity model, designed based on domain expertise. We demonstrate that for four benchmark tasks and two computationally expensive neuroscience simulators, our multifidelity approach can identify the posterior distributions more efficiently than NPE and TSNPE, often reducing the number of required high-fidelity simulations by orders of magnitude.

## 2 BACKGROUND

**Multifidelity methods for inference**  Multifidelity has been widely explored in the context of likelihood-based inference (Peherstorfer et al., 2018), from maximum likelihood estimation approaches (Maurais et al., 2023) to Bayesian inference methods (Vo et al., 2019; Catanach et al., 2020). For cases where the likelihood is not explicitly available, several sampling-based multifidelity methods have been proposed within the framework of ABC (Prescott & Baker, 2020; Warne et al., 2022; Prescott et al., 2024; Prescott & Baker, 2021). However, these methods inherit limitations of ABC approaches, particularly in high-dimensional parameter spaces, where neural density estimators offer more scalable alternatives to complex real-world problems (Lueckmann et al., 2021). Concurrently with our work, Thiele et al. (2025) developed a multifidelity SBI approach based on response distillation, Hikida et al. (2025) adapted multilevel Monte Carlo techniques to SBI, and Saoulis et al. (2025) applied transfer learning to accelerate inference on a cosmological task.

Beyond SBI, multifidelity has been explored in Bayesian optimization, where Gaussian process models integrate data of different fidelities to infer expensive functions (e.g., Song et al., 2019; Zanjani Foumani et al., 2023). These approaches focus on learning surrogate likelihood functions rather than posteriors over simulator parameters, but they highlight the broad applicability of the multifidelity concept.

**Transfer learning and simulators** To facilitate learning in a target domain, transfer learning borrows knowledge from a source domain (Panigrahi et al., 2021). This is often done when the target dataset is smaller than the source dataset (Larsen-Freeman, 2013). For numerical simulators, transfer learning approaches have been used to lower the simulation budget, for instance, in $CO_2$ forecasting (Falola et al., 2023), surrogate modeling (Wang et al., 2024; Zeng et al., 2026), and model inversion with physics-informed neural networks (Haghighat et al., 2021). To the best of our knowledge, the potential of transfer learning for computationally efficient simulation-based inference has not been fully realized yet.

**Simulation-efficient SBI** Recent work reduces the cost of SBI for expensive simulators through active learning or efficient representations. Active learning methods adaptively select simulation parameters for neural likelihood or posterior estimation (Lueckmann et al., 2019; Griesemer et al., 2024), paralleling Bayesian optimization for ABC (Gutmann & Corander, 2016). Efficiency also improves through learned representations such as signature-based features (Dyer et al., 2022), compositional models (Gloeckler et al., 2025), or self-consistency objectives (Schmitt et al., 2024a;b). Unlike these single-fidelity approaches, MF-(TS)NPE leverages an expert-designed low-fidelity simulator and combines transfer learning with active learning to refine posterior estimates efficiently.

## 3 METHODS

MF-(TS)NPE is a multifidelity approach to Neural Posterior Estimation (NPE) for computationally expensive simulators leveraging transfer learning and, in its sequential variant, active learning. We present our approach in Sec. 3.1. In Sec. 3.1.4, we discuss the evaluation metrics used to compare our method against NPE (Greenberg et al., 2019), TSNPE (Deistler et al., 2022), and MF-ABC (Prescott & Baker, 2020). MF-(TS)NPE is summarized in Fig. 1, Algorithms 1 and 3.

### 3.1 MULTIFIDELITY NPE

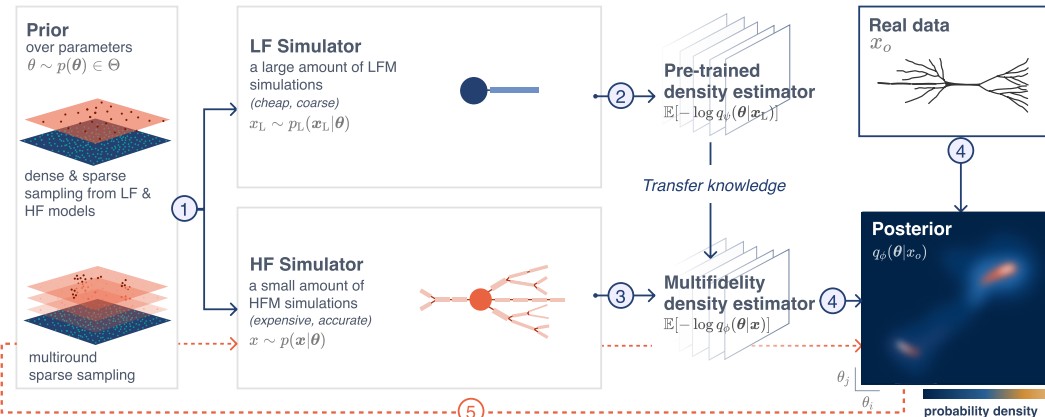

Figure 1: **Multifidelity Neural Posterior Estimation** proceeds by dense sampling from the prior distribution, running the low-fidelity simulator (*e.g., a two-compartment neuron model* (Hodgkin & Huxley, 1952)), and training a neural density estimator with a negative log-likelihood loss. MF-NPE then retrains the pre-trained network on sparse samples from the same prior distribution and respective high-fidelity simulations (*e.g., a multicompartmental neuron model* (Rall, 1995)). Given empirical observations $x_o$, MF-NPE estimates the posterior distribution given the high-fidelity model. In the sequential case, the parameters for high-fidelity simulations are drawn from iterative refinements of the prior distribution within the support of the current posterior estimate, at some observation $x_o$.

We aim to infer the posterior distribution over the parameters $\theta$ of a computationally expensive high-fidelity simulator $p(x|\theta)$, with computational cost of a single simulation $c$. We designate the simulator as high-fidelity if the model accurately captures the empirical phenomenon, but incurs high computational cost when generating simulations. We assume that we have access to a low-fidelity

simulator $p_L(\boldsymbol{x}_L|\boldsymbol{\theta})$, describing a simplification of the phenomenon of interest with cost $c_L \ll c$. We assume that both simulators operate over the same domain of observations $\boldsymbol{x}$, and the parameters of the low-fidelity model form at least a subset (and at most the entirety) of the high-fidelity parameters. Our goal is to develop an estimator that leverages low-fidelity simulations to infer the posterior distribution over parameters of the high-fidelity model with limited high-fidelity simulations, without access to a tractable likelihood for either simulator.

As with NPE (Papamakarios & Murray, 2016; Greenberg et al., 2019), to estimate the posterior density over model parameters $\boldsymbol{\theta}$ for which the likelihood function is unavailable, we consider a sufficiently expressive neural density estimator $q_\phi(\boldsymbol{\theta}|\boldsymbol{x})$, and train it to minimize the negative log-likelihood loss:

$$\mathcal{L}(\phi) = \mathbb{E}_{\theta \sim p(\boldsymbol{\theta})} \mathbb{E}_{x \sim p(\boldsymbol{x}|\boldsymbol{\theta})} \left[ -\log q_\phi(\boldsymbol{\theta}|\boldsymbol{x}) \right], \tag{1}$$

where $\boldsymbol{\theta}$ is sampled from the prior distribution, $\boldsymbol{x}$ denotes the respective simulations (i.e., samples from $p(\boldsymbol{x}|\boldsymbol{\theta})$), and $\phi$ are the network parameters. By minimizing $\mathcal{L}(\cdot)$, the neural density estimator approximates the conditional distribution $p(\boldsymbol{\theta}|\boldsymbol{x})$ directly (Papamakarios & Murray, 2016) (proof of convergence in Appendix B). Given an empirical observation $\boldsymbol{x_o}$, we can then estimate the posterior over parameters $p(\boldsymbol{\theta}|\boldsymbol{x_o})$. To ensure $q_\phi(\boldsymbol{\theta}|\boldsymbol{x_o})$ closely approximates the true posterior $p(\boldsymbol{\theta}|\boldsymbol{x_o})$, the density estimator must be sufficiently expressive. We use neural spline flows (NSFs) (Durkan et al., 2019), expressive normalizing flows that have been shown empirically to be competitive for SBI (Lueckmann et al., 2021). To avoid overfitting when training NSFs, we use the same validation-based early stopping criterion $S$ as in the SBI package (Boelts et al., 2024) (details in Appendix.C.1).

### 3.1.1 TRANSFER LEARNING

MF-NPE leverages representations learned from low-fidelity simulations to reduce the number of high-fidelity simulations required to approximate a high-fidelity posterior. To that end, MF-NPE adopts a *fine-tuning* strategy of transfer learning: Let $\psi$ be the parameters of the low-fidelity neural density estimator $q_\psi(\boldsymbol{\theta}|\boldsymbol{x}_L)$ and let $\phi$ be the parameters of the high-fidelity density estimator $q_\phi(\boldsymbol{\theta}|\boldsymbol{x})$. MF-NPE minimizes the loss $\mathcal{L}(\phi) = \mathbb{E}_{\theta \sim p(\boldsymbol{\theta})} \mathbb{E}_{x \sim p(\boldsymbol{x}|\boldsymbol{\theta})} \left[ -\log q_\phi(\boldsymbol{\theta}|\boldsymbol{x}) \right]$ on the high-fidelity task, where the parameters $\phi$ are initialized on the pretrained low-fidelity network parameters $\psi$. We argue that by pre-training on low-fidelity simulations, the density estimator learns useful features up front (i.e., the feature spaces of the low- and high-fidelity density estimators overlap), so fewer high-fidelity simulations suffice to refine the posterior estimates. Indeed, Tahir et al. (2024) shows that once networks learn suitable features for a given predictive task, they drastically reduce the sample complexity for related tasks. Other strategies to pretraining are discussed in Appendix G.4.

MF-NPE can naturally accommodate more than two fidelity levels (Appendix L), does not require more hyperparameter tuning than NPE (Appendix C.1), and is applicable in situations where the low-fidelity model has fewer parameters than the high-fidelity model. In this setting, the parameters that are exclusive to the high-fidelity model are treated as dummy variables in the pre-trained density estimator. The pre-conditioning with these variables leads to the pre-trained neural density estimator to effectively estimate the prior distribution over the respective parameters (OU3 and OU4 in Appendix I.1). As shown below, our method is compatible with both embedding networks and hand-crafted summary statistics of the observations.

---

**Algorithm 1** MF-NPE

---

1: **Input:** $N$ pairs of $(\boldsymbol{\theta}, \boldsymbol{x}_L)$; $M$ pairs of $(\boldsymbol{\theta}, \boldsymbol{x})$; conditional density estimators $q_\psi(\boldsymbol{\theta}|\boldsymbol{x}_L)$ and $q_\phi(\boldsymbol{\theta}|\boldsymbol{x})$ with respectively learnable parameters $\psi$ and $\phi$; early stopping criterion $S$.
2: $\mathcal{L}(\psi) = \frac{1}{N} \sum_{i=1}^{N} -\log q_\psi \left( \boldsymbol{\theta}_i | \boldsymbol{x}_i^L \right)$.    **/\* Low-fidelity model \*/**
3: **for** epoch in epochs **do**
4:     train $q_\psi$ to minimize $\mathcal{L}(\psi)$ until $S$ is reached.
5: **end for**
6: Initialize $q_\phi$ with weights and biases of trained $q_\psi$.    **/\* High-fidelity model \*/**
7: $\mathcal{L}(\phi) = \frac{1}{M} \sum_{i=1}^{M} -\log q_\phi \left( \boldsymbol{\theta}_i | \boldsymbol{x}_i \right)$.
8: **for** epoch in epochs **do**
9:     train $q_\phi$ to minimize $\mathcal{L}(\phi)$ until $S$ is reached.
10: **end for**

---

### 3.1.2 SEQUENTIAL TRAINING

In addition to learning amortized posterior estimates with NPE, our approach naturally extends to sequential training schemes when estimating the non-amortized posterior $q_\phi(\boldsymbol{\theta}|\boldsymbol{x_o})$. Rather than sampling model parameters from the prior, sequential methods introduce an active learning scheme that iteratively refines the posterior estimate for a specific observation $\boldsymbol{x_o}$. These methods – known as Sequential Neural Posterior Estimation (Papamakarios & Murray, 2016; Lueckmann et al., 2017) – have shown increased simulation efficiency when compared to NPE (Lueckmann et al., 2021). However, applying these methods with flexible neural density estimators requires a modified loss that suffers from instabilities in training and posterior leakage (Greenberg et al., 2019). Truncated Sequential Neural Posterior Estimation (TSNPE) mitigates these issues by sampling from a truncated prior distribution that covers the support of the posterior. This leads to a simplified loss function and increased training stability, while retaining performance (Deistler et al., 2022).

We apply our multifidelity approach to TSNPE. First, the high-fidelity density estimator is initialized from the learned network parameters of a low-fidelity density estimator. Then, high-fidelity simulations are generated iteratively from a truncated prior, within the support of the current posterior. We refer to this method as MF-TSNPE (complete description of the algorithm in Appendix M.1).

### 3.1.3 ACQUISITION FUNCTION

To further enhance the efficiency of our sequential algorithm, we explore the use of acquisition functions to supplement our round-wise samples from the TSNPE proposal: we generate simulations for round $i$ with a set of parameters $\boldsymbol{\theta}^{(i)} = \{\boldsymbol{\theta}^{(i)}_{\text{prop}} \cup \boldsymbol{\theta}^{(i)}_{\text{active}}\}$ where $\boldsymbol{\theta}^{(i)}_{\text{prop}}$ are samples from the proposal distribution at round $i$, and $\boldsymbol{\theta}^{(i)}_{\text{active}}$ are the top $\mathcal{B}$ values according to an acquisition function. We refer to this algorithm as MF-TSNPE-AF (full description in Appendix M.2). Following Järvenpää et al. (2019); Lueckmann et al. (2019), we select an acquisition function that targets the variance of the posterior estimate with respect to the epistemic uncertainty in the learned parameters $\phi|\mathcal{D}$.

$$\boldsymbol{\theta}^* = \underset{\boldsymbol{\theta}}{\arg\max} \mathbb{V}_{\phi|\mathcal{D}}[q_\phi(\boldsymbol{\theta}|\boldsymbol{x_o})] \tag{2}$$

We realize this as the sample variance across an ensemble of neural density estimators trained independently on the same dataset $\mathcal{D}$, as done in Lueckmann et al. (2019). Note that we use epistemic uncertainty to guide high-fidelity simulation selection within the simulator's domain rather than out-of-distribution samples. For details on the proposal design of MF-TSNPE-AF, see Appendix M.2.

### 3.1.4 EVALUATION METRICS

We evaluate the method on observations $\boldsymbol{x_o}$ from the high-fidelity simulator, with parameter values drawn from the prior distribution. This ensured a fair evaluation of how much the low-fidelity simulator helps to infer the posterior distribution given the high-fidelity model. All methods were evaluated for a range of high-fidelity simulation budgets $(50, 10^2, 10^3, 10^4, 10^5)$, on posteriors given the same data set of observations $\boldsymbol{x_o}$.

**Known true posterior** We evaluate the accuracy of posterior distributions in cases where the ground-truth posterior is known with the Classifier-2-Sample Test (C2ST) and the Maximum Mean Discrepancy (MMD)(Friedman, 2004; Lopez-Paz & Oquab, 2017; Gretton et al., 2012; Lueckmann et al., 2021; Peyré & Cuturi, 2017). C2ST is commonly used in SBI, as it is easy to apply and interpret: a value close to 0.5 means that a classifier cannot effectively distinguish the two distributions, implying the posterior estimate is close to the ground-truth posterior. A value close to 1 means that the classifier can distinguish the distributions very well, indicating a poor posterior estimation. C2ST is rarely applicable in practical SBI settings, since it requires samples from the true posterior (e.g., Sec. 4.1).

**Unknown true posterior** The average Negative Log probability of the True Parameters (NLTP; $-\mathbb{E}[\log q(\boldsymbol{\theta_o}|\boldsymbol{x_o})]$ ) has been extensively used in the SBI literature for problems where the true posterior is unknown (Greenberg et al., 2019; Papamakarios & Murray, 2016; Durkan et al., 2020; Hermans et al., 2020). In the limit of a large number of pairs $(\boldsymbol{\theta_o}, \boldsymbol{x_o})$, the average over the log probability of each pair $(\boldsymbol{\theta_o}, \boldsymbol{x_o})$ approaches the expected KL divergence between the estimated

and the true posterior (up to a term that is independent of the estimated posterior), as shown in (Lueckmann et al., 2021). In addition, we report the Normalized Root Mean Square Error (NRMSE), which quantifies the deviation of posterior samples from the true parameters on a scale-invariant axis. NRMSE values closer to 0 indicate better predictive performance.

# 4 RESULTS

We evaluate the performance of our multifidelity approach to NPE and TSNPE on six tasks involving various types of observations (e.g., time series, images, neural spiking). We start with four benchmarking tasks, followed by two challenging neuroscience problems with computationally expensive simulators and for which no likelihood is available: a multicompartmental neuron model and a neural network model with synaptic plasticity. We also provide a comparison to MF-ABC (Sec. E.1.1, D.3). In Sec. 4.4, we provide a discussion about the effectiveness of transfer learning in MF-NPE.

## 4.1 BENCHMARKING TASKS

We first evaluated MF-(TS)NPE on four benchmarking tasks: **SIR**, **SLCP**, **OUprocess**, and **Gaussian Blob**. SIR and SLCP are established SBI benchmarks (Lueckmann et al., 2021), OUprocess is a new multifidelity task with tractable likelihood (Kou et al., 2012), and Gaussian Blob is a high-dimensional image task (Lueckmann et al., 2019) (details in Appendix D). These tasks were chosen to systematically investigate various task properties that might impact the performance of transfer learning in a multifidelity setting: differing parameter dimensionality between the low- and high-fidelity models, partly observed dynamics, differing simulator types between the low- and high-fidelity models, and high-dimensional observations. Furthermore, these multifidelity tasks are not trivial in the sense that the low and high-fidelity simulators lead to different posteriors (Appendix I). Note that we do not evaluate the total cost of low- and high-fidelity simulations in these tasks, but defer this analysis to the two complex neuroscience tasks (Appendix J).

To evaluate MF-NPE, we compared the estimated densities to the respective reference posterior, estimated from the exact likelihood with Rejection Sampling (Martino et al., 2018) (OU process; closed-form of the likelihood in Sec. D.1), and using Sampling and Importance Resampling (RUBIN, 1988) to obtain a set of 10k proposal samples (SLCP, SIR), similar to Lueckmann et al. (2021). We quantified the performance with C2ST and MMD over 10 observations (30 observations for the OU process) and 10 network initializations per observation. GaussianBlob required a CNN embedding and was evaluated with NRMSE and NLTP since no closed-form likelihood is available (Fig. 11).

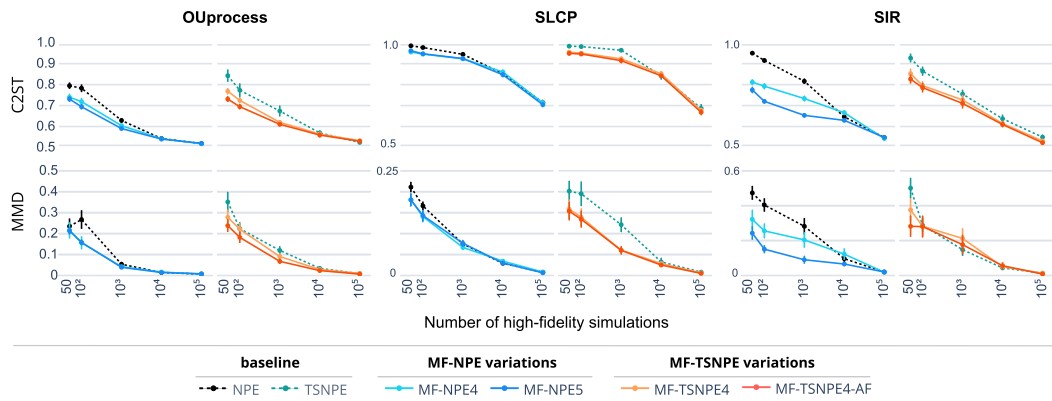

Figure 2: C2ST and MMD averaged over 10 network initializations with means and $95\%$ confidence intervals. MF-NPE4 and MF-NPE5 are pretrained on $10^4$ and $10^5$ low-fidelity simulations, respectively. Results for the GaussianBlob task in Fig. 11; variations on the OU task and comparisons to MF-ABC in Fig. 8.

Across four benchmarking tasks, we observed a consistent performance increase with MF-NPE compared to NPE, and MF-TSNPE(-AF) compared to TSNPE, especially in low simulation budgets from the high-fidelity model ($50$-$10^3$ simulations) (Fig. 2; Gaussian Blob in Fig. 11). In addition,

we found that having a higher number of low-fidelity samples improved performance, reinforcing that low-fidelity simulations were indeed advantageous for pre-training the neural density estimator for the downstream task. Note that for the OU and SLCP tasks, we did not observe a substantial increase in MF-NPE performance between the settings with $10^4$ and $10^5$ low-fidelity samples, suggesting an upper bound regarding pre-training efficacy. We also compared MF-NPE with MF-ABC, an ABC-based method for multifidelity SBI (Prescott & Baker, 2020), and observed that MF-NPE has a substantially higher performance (Appendix E.1.1). This is consistent with previous findings indicating the superior performance of NPE with respect to rejection ABC and SMC-ABC, where it is not uncommon to require orders of magnitude more simulations to obtain reliable posterior approximations (Lueckmann et al., 2021; Frazier et al., 2024). However, a more extensive hyperparameter search could potentially lead to substantial improvements in MF-ABC performance.

As described in Sec. 3, we enhanced the sequential algorithm TSNPE (Deistler et al., 2022) with a first round of MF-NPE, and designated this approach as MF-TSNPE. We found that MF-TSNPE (details in Appendix M.1) performs better than TSNPE, especially in regimes with a low budget of high-fidelity simulations. Compared to MF-TSNPE, MF-TSNPE-AF improved inference in the OU process, but did not show significant improvements in the SLCP and SIR tasks.

Finally, we assessed the contribution of transfer learning to the overall performance in a setting where the low- and high-fidelity models have a different number of parameters, in the context of the OUprocess task (Appendix D.3). We expected that adding parameters to the high-fidelity model that are absent in the low-fidelity model would increase the inference complexity for MF-NPE, and indeed observed a performance decrease in MF-NPE, although MF-NPE still performed better than NPE and MF-ABC (see Appendix D.3). We note that MF-NPE also outperformed NPE when the low-fidelity model had more parameters than the high-fidelity model (see Appendix D.4). Overall, the results suggest that MF-NPE and MF-TSNPE can yield substantial performance gains compared to NPE, TSNPE, and MF-ABC.

## 4.2 MULTICOMPARTMENTAL NEURON MODEL

The voltage response of a morphologically-detailed neuron to an input current is typically modeled with a multicompartment model wherein the voltage dynamics of each compartment are based on the Hodgkin-Huxley equations (Hodgkin & Huxley, 1952). The higher the number of compartments of the model, the more accurate the model is, but the higher the simulation cost.

In this task, we aimed to infer the densities of ion channels $\bar{g}_{Na}$ and $\bar{g}_K$ on a morphologically-detailed model of a thick-tufted layer 5 pyramidal cell (L5PC) containing 8 compartments per branch (Fig. 3A) (Van Geit et al., 2016). We injected in the first neuron compartment a noisy 100 ms step current with mean $I_m = 0.3$ nA: $I_e = I_m + \epsilon, \epsilon \sim \mathcal{N}(0, 0.01)$. The voltage response of the neuron was recorded over 120 ms, with a simulation step size of 0.025 ms and 10 ms margin before and after the current injection. We defined the high-fidelity model to have 8 compartments per branch and the low-fidelity model to have 1 compartment per branch, and both the high and low-fidelity models had the same injected current and ion channel types.

To simulate the neuron models, we used Jaxley, a Python toolbox for efficiently simulating multicompartment single neurons with biophysical detail (Deistler et al., 2025). In this setting, the simulation time for the high-fidelity model is approximately 4 times higher than that of the low-fidelity model. We characterized the neural response with four summary statistics that have been commonly used when fitting biophysical models of single neurons to empirical data: spike count, mean resting potential, standard deviation of the resting potential, and voltage mean (Gonçalves et al., 2020; Gao et al., 2023). Performances were evaluated with NLTP and NRMSE on $10^3$ pairs of $\theta_o$ and respective simulation outputs $x_o$, averaged over 10 random network initializations (Sec. 3.1.4).

MF-(TS)NPE showed higher performance than NPE, in particular with larger low-fidelity simulation budgets (Fig. 3B; Fig. F.1), despite the right-skewed posterior distribution of the low-fidelity model (Fig. 21). Furthermore, MF-NPE posterior predictives closely matched the empirical data, in contrast with NPE, even when NPE was trained on a higher number of high-fidelity simulations (Appendix F). In addition, MF-(TS)NPE achieved comparable performance with a total computational cost $4.44 \pm 0.06$ times lower than standard NPE (Appendix J). Finally, TARP and simulation-based calibration tests suggest that both MF-NPE and NPE estimates were relatively well calibrated (Fig. 3C) (Talts et al., 2020; Lemos et al., 2023).

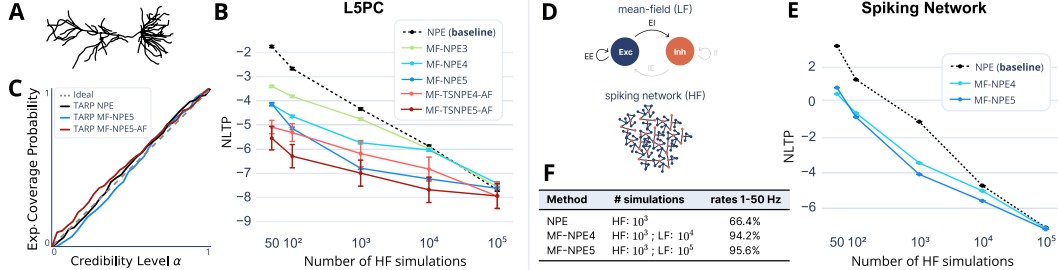

Figure 3: **(A)** Thick-tufted layer 5 pyramidal cell from the neocortex. **(B)** Performance evaluation with NLTP (same naming convention as in Fig. 2). Amortized methods are averaged over 10 network initializations; non-amortized trained once per 100 observations. Similar results were obtained with NRMSE (Appendix F.1). MF-NPE, and especially its sequential variants, are orders of magnitude more simulation-efficient than NPE. **(C)** TARP posterior calibration check shows that NPE and MF-NPE trained on $10^3$ high-fidelity samples are well-calibrated (Lemos et al., 2023). Simulation-based calibration, posterior samples, and predictives are in Appendix F. **(D)** Schematic of the low and high-fidelity models of a spiking network. **(E)** Performance of NPE and MF-NPE evaluated on 10000 true observations with NLTP: averages over 10 network initializations, and 95% confidence intervals. **(F)** Proportion of posterior samples within the target firing rate bounds. MF-NPE produces a higher fraction of parameter sets within the bounds than NPE.

MF-TSNPE-AF pre-trained on $10^4$ low-fidelity samples outperforms MF-NPE trained on $10^5$ samples. However, MF-TSNPE-AF performance comes at the cost of training time due to the use of an ensemble of density estimators (Appendix J). This additional training burden is only justified when the simulation cost is substantially higher than the training cost.

## 4.3 Recurrent spiking network

Finally, we applied MF-NPE to a challenging and timely problem in neuroscience: the inference of synaptic plasticity rules that endow large spiking neural networks with dynamics reminiscent of experimental data. This problem has been recently tackled with an SBI method (filter simulation-based inference, fSBI) that progressively narrows down the search space of parameters given different sets of summary statistics (Confavreux et al., 2023). fSBI was successful in obtaining manifolds of plasticity rules that ensure plausible network activity, but the compute requirements were reported to be very large. Here, we aim to test whether this problem can be efficiently tackled with MF-NPE.

The high-fidelity simulator consisted of a recurrent network of 4096 excitatory ($E$) and 1024 inhibitory ($I$) leaky integrate-and-fire neurons connected with conductance-based synapses (Fig. 3D). Each synapse type in this network ($E$-to-$E$, $E$-to-$I$, $I$-to-$E$, $I$-to-$I$) was plastic with an unsupervised local learning rule. For each synapse type, 6 parameters governed how the recent pre- and post-synaptic activity were used to update the synapse, for a total of 24 free parameters across all 4 synapse types (Confavreux et al., 2023). The networks were simulated using Auryn, a C++ simulator (Zenke & Gerstner, 2014) (details in Appendix G).

Mean-field theory can be applied to the dynamical system above to obtain the steady-state activities of the excitatory and inhibitory populations as a function of the parameters of the plasticity rules embedded in the network. Though such analysis is widely performed in the field (Vogels et al., 2011; Confavreux et al., 2023; Gerstner et al., 2014), it has never been used as a low-fidelity model to help with the inference of the high-fidelity model parameters. Since there are no dynamics to simulate with the mean-field model, the simulation was almost instantaneous, while the high-fidelity model took approximately 5 minutes to generate a single 2-minute long simulation on a single CPU.

Summary statistics of the low- and high-fidelity models were the average firing rates of the excitatory and inhibitory neurons at steady state (after 2 minutes of simulation in the high-fidelity model). Plastic networks were considered plausible if the firing rates were between 1 and 50Hz (Dayan & Abbott, 2001; Confavreux et al., 2023).

In this task, the low-fidelity model focuses solely on the $E$-to-$E$ and $E$-to-$I$ rules from the high-fidelity model, thereby having 12 out of the 24 parameters of the high-fidelity model. This setup allows us to demonstrate the performance of MF-NPE on problems with different parameter spaces, highlighting MF-NPE's flexibility and advantages. We found that MF-NPE has better performance than NPE in terms of NLTP (Fig. 3E), although we observed a diminishing performance gain with increasing discrepancy between the number of parameters of the low- and high-fidelity models (see Appendix G.3). Furthermore, MF-NPE leads to an increase of almost $30\%$ in the proportion of posterior samples within the target firing rate bounds (Fig. 3F), reinforcing that MF-NPE is a practical and effective method for SBI of costly real-world simulators.

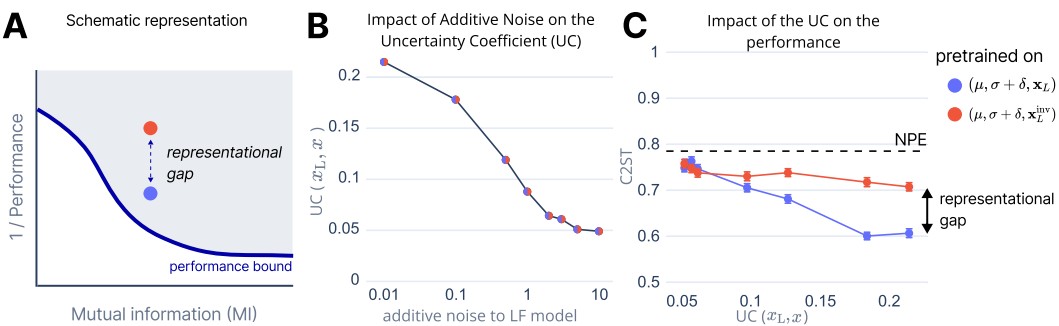

Figure 4: **(A)** Schematic figure representing lower bound on transfer error ($1/$MF-NPE performance) as a function of mutual information between the low- and high-fidelity models, given a fixed simulation budget. **(B)** Uncertainty coefficient monotonically decreases with noise parameter $\delta$ and is invariant to data inversion. **(C)** Empirical results with MF-NPE support the hypothesis that transfer performance is dependent on both mutual information and representational coherence. Note that NPE (with the same high-fidelity simulation budget of $10^2$) has similar performance as MF-NPE in the case where the low- and high-fidelity models have low mutual information.

## 4.4 WHEN DOES PRE-TRAINING HELP?

In previous sections, we demonstrated that MF-NPE can significantly reduce the number of high-fidelity simulations required to accurately approximate the high-fidelity posterior by leveraging pre-training on low-fidelity simulations. This naturally leads to several key questions: Which characteristics of low-fidelity simulators enable effective transfer learning? Under what conditions can pre-training reliably enhance simulation efficiency?

Providing theoretical guarantees for these questions necessitates a formal characterization of convergence rates in NPE with transfer learning. Although recent works have begun addressing these challenges in NPE (Frazier et al., 2024), current theoretical frameworks of transfer learning (Tahir et al., 2024; Yun et al., 2020; Tripuraneni et al., 2020; Lampinen & Ganguli, 2018), rely on simplifying assumptions (e.g., linear networks) that do not fully capture the complexities of MF-NPE. Given this limitation, we instead empirically explored the conditions in which low-fidelity pre-training facilitates effective transfer learning. To do this, we evaluated MF-NPE where the low- and high-fidelity simulators were related by systematic perturbations (Fig. 4).

We hypothesized that the effectiveness of pre-training is associated with two primary factors:

1. **Mutual information** between the low- and high-fidelity simulators.

2. **Representational coherence**, i.e., similarity in how task-relevant information is encoded.

To isolate the effects of these factors, we constructed controlled variants of the OU2 process in which the low-fidelity simulator differs from the high-fidelity one through two distinct transformations. In the baseline setup, the simulators generate observations according to

$$x \sim p(x \mid \mu, \sigma), \qquad x_{\mathrm{L}} \sim p(x \mid \mu, \sigma + \delta),$$

where the perturbation $\delta$ increases the noise of the low-fidelity simulator and therefore reduces $\mathbb{I}[x; x_{\mathrm{L}}]$ monotonically as $\delta$ grows.

Second, to independently manipulate representational coherence, we applied an invertible coordinate-reversal transformation $x_{\mathrm{L}}^{\mathrm{inv}} = T(x_{\mathrm{L}})$, implemented via an anti-diagonal permutation matrix that reverses the ordering of the output dimensions. Because $T$ is invertible, the mutual information between the two simulators is unchanged:

$$\mathbb{I}[\boldsymbol{x};\, \boldsymbol{x}_{\mathrm{L}}^{\mathrm{inv}}] \;=\; \mathbb{I}[\boldsymbol{x};\, \boldsymbol{x}_{\mathrm{L}}] \;=\; \mathbb{H}[\boldsymbol{x}] + \mathbb{H}[\boldsymbol{x}_{\mathrm{L}}] - \mathbb{H}[\boldsymbol{x}, \boldsymbol{x}_{\mathrm{L}}].$$

Thus, while $\mathbb{I}[x; x_{\mathrm{L}}]$ decreases monotonically with the noise scale $\delta$, the inversion leaves the information content unchanged while disrupting representational coherence. Figure 4 illustrates how each manipulation affects the uncertainty coefficient (Figure 4B), which we estimated empirically using MINE (Belghazi et al., 2018), and MF-NPE performance under a fixed simulation budget of $10^4$ low-fidelity and $10^2$ high-fidelity simulations (Figure 4C).

In agreement with our hypothesis, our results suggest that the effectiveness of MF-NPE depends on both the mutual information and the representational coherence between low- and high-fidelity simulators (Fig. 4C). Specifically, mutual information is necessary for effective transfer learning but not sufficient: perturbations that preserve information (e.g., invertible transformations) can still substantially impair transfer performance. Effective pre-training strategies should therefore prioritize low-fidelity simulators that are both highly informative and representationally aligned with the high-fidelity model.

## 5 DISCUSSION

We proposed a new method for simulation-based inference that leverages low-fidelity models to efficiently infer the parameters of costly high-fidelity models. By incorporating transfer learning and multifidelity approaches, MF-NPE substantially reduces the simulation budget required for accurate posterior inference. This addresses a pervasive challenge across scientific domains: the high computational cost of simulating complex high-fidelity models and linking them to empirical data. Our empirical results demonstrate MF-NPE's competitive performance in SBI across statistical benchmarks and real-world applications, as compared to a standard method such as NPE.

**Limitations** Despite MF-NPE's advantages, the method comes with some challenges. First, the effectiveness of MF-NPE relies on the similarity between the low-fidelity and high-fidelity models. Fortunately, in many situations, domain experts will know beforehand whether low-fidelity models are poor approximations of high-fidelity models. Second, MF-NPE and MF-TSNPE inherit the limitations of NPE and TSNPE, respectively, in particular regarding the scalability of simulation-based inference to high-dimensional parameter spaces. How to balance exploration of high-dimensional parameter spaces and computational cost in a simulation-based inference setting remains a topic of active research. Third, MF-TSNPE-AF requires the training of an ensemble of density estimators, which leads to substantial computational costs in training and hyperparameter tuning. This method should therefore only be preferred in cases where the cost incurred in simulations outweighs the training cost. We estimate this to be the case for the tasks with the multicompartment neuron model and the spiking network model, for which the cost of one simulation and the training of one density estimator are comparable in certain settings (e.g., on the order of minutes, for a network trained on $10^3$ samples).

**Future work** We identify three promising research directions for multifidelity simulation-based inference. First, we expect the scalability and expressivity of MF-NPE could be improved by utilizing the same approaches of multifidelity and transfer learning presented here with neural density estimators other than normalizing flows, such as diffusion models (Gloeckler et al., 2024). Second, we assumed a negligible cost for low-fidelity simulations, and future work should address how to optimally allocate low- and high-fidelity simulations under a fixed computational budget. Third, similar to past efforts in developing a benchmark for simulation-based inference, it will be beneficial for the SBI community to develop a benchmark for multifidelity problems, with new tasks, algorithms and evaluation metrics. This will promote rigorous and reproducible research and catalyze new developments in multifidelity SBI, and in SBI more generally. Our work and codebase are a step in this direction.

**Conclusion** Overall, MF-(TS)NPE is a method for simulation-based inference that leverages low-fidelity models and transfer learning to infer the parameters of costly high-fidelity models, thus providing an effective balance between computational cost and inference accuracy.

## 6 REPRODUCIBILITY STATEMENT

The training and simulation costs for all tasks and SBI methods, as well as a detailed description of the experimental setup, are described in Appendices C.1 and J. The corresponding code and data are publicly available on Github: github.com/goncalab/multifidelity-NPE.

### ACKNOWLEDGMENTS

We thank Karthik Sama, Najlaa Mohamed, Guy Moss, Marcel Nonnenmacher and Pierre Vanvolsem for discussions. We also thank the staff of the VIB Data Core for their support. Anastasia N. Krouglova was supported by an FWO grant (G097022N). Hayden R. Johnson was supported by an FWO grant (G053624N). Basile Confavreux was supported by a Schmidt Science Polymath Award to Andrew Saxe, the Sainsbury Wellcome Centre Core Grant from Wellcome (219627/Z/19/Z) and the Gatsby Charitable Foundation (GAT3850). Michael Deistler was supported by the German Research Foundation (DFG) through Germany's Excellence Strategy (EXC 2064 – Project number 390727645), the German Federal Ministry of Education and Research (Tübingen AI Center, FKZ: 01IS18039A) and the European Union (ERC, "DeepCoMechTome", ref. 101089288; Jakob H. Macke). Views and opinions expressed are however those of the authors only and do not necessarily reflect those of the European Union or the European Research Council Executive Agency. Neither the European Union nor the granting authority can be held responsible for them. Michael Deistler was a member of the International Max Planck Research School for Intelligent Systems (IMPRS-IS). Pedro J. Gonçalves is thankful for the financial support from the Flemish government through long-term structural funding Methusalem (grant METH/26/003).

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

## A    USAGE OF LLMS

LLM usage was minimal, limited to grammar refinement, sentence shortening, code cleanup and discovering papers outside our main domain.

## B    PROOF OF CONVERGENCE OF THE NPE LOG-LIKELIHOOD LOSS

Let $\theta_i \sim p(\theta_i)$ be samples from the prior of a high-fidelity model, and $x_i \sim p(x|\theta_i)$ be the respective high-fidelity simulations. In NPE, we define the loss function as the negative log likelihood:

$$\mathcal{L}(\phi) = -\frac{1}{N} \sum_i^N \log q_\phi(\theta_i|x_i), \tag{3}$$

where $\theta_i$ are samples from the prior distribution, $x_i$ are the respective simulations (i.e., samples from $p(x|\theta_i)$), and $\phi$ are the parameters of the neural density estimator to be optimized. If we let the number of samples $\theta_i$ (and respective simulations) $N \to \infty$:

$$
\begin{aligned}
\mathcal{L}(\phi) &= \mathbb{E}_{p(\theta)p(x|\theta)} \left[ -\log q_\phi(\theta|x) \right] \\
&= \mathbb{E}_{p(x)p(\theta|x)} \left[ -\log q_\phi(\theta|x) \right] \\
&= \mathbb{E}_{p(x)} \left[ \mathbb{E}_{p(\theta|x)} \left[ \log \frac{p(\theta|x)}{q_\phi(\theta|x)} \right] \right] + C \\
&= \mathbb{E}_{p(x)} [D_{KL} \left( p(\theta|x), q_\phi(\theta|x) \right)] + C
\end{aligned}
\tag{4}
$$

where $C$ is a constant with respect to $\phi$. Minimizing $\mathcal{L}(\phi)$ with respect to $\phi$ is thus equivalent to minimizing the KL divergence between the true posterior distribution and the estimated posterior in the limit of an infinite number of high-fidelity samples.

## C  FURTHER EXPERIMENTAL DETAILS

### C.1  TRAINING PROCEDURE

All methods and evaluations were implemented in PyTorch (Paszke et al., 2019). We used the Zuko package (version 1.4.0, MIT License)[1] (Roset, 2024) to implement the normalizing flow, based on the Neural Spline Flows (NSF) architecture (Durkan et al., 2019), and the SBI package (version 0.24.0, Apache 2.0 license)[2] (Boelts et al., 2024) for additional functions. The parameters used to generate simulations were logit-transformed for numerical stability, and the summary statistics were z-scored to improve the performance of the normalizing flows. The loss function is the negative-log likelihood, and the optimization function is the *Adam optimizer* (Kingma & Ba, 2017).

The Neural Spline Flow (NSF) architecture consists of 5 transformations, each parametrized with 50 hidden units and 8 bins. The batch size was set to 200, and the learning rate to $5 \times 10^{-4}$. The train-validation fraction is $0.1$, and training of the NSF utilized an early stopping criterion with a patience of 20 epochs for the early stopping criterion. The settings described above are all default settings of the SBI package at the time of the method's development (Boelts et al., 2024).

Note, the stopping criterion follows the default configuration of the SBI package, which is defined as follows: Let $E$ be the error function of the training algorithm (negative log likelihood), $E_{val}(t)$ the validation error at epoch $t$, which is used by the stopping criterion. The value $E_{opt}(t)$ is the lowest validation set error obtained in epochs up to $t$:

$$E_{opt}(t) := \min_{t' \le t} E_{val}(t') \tag{5}$$

The early stopping criterion $S$ terminates training once the validation error $E_{\mathrm{val}}(t)$ has increased for $p$ consecutive epochs (the patience parameter). At this point, the model corresponding to the lowest validation error observed that far, $E_{\mathrm{opt}}(t)$, is selected and returned.

Rather than fixing the number of training epochs, the idea behind early stopping is that when the validation error has increased not only once, but over $p$ consecutive steps, such an increase indicates a stage of overfitting (Prechelt, 2002). Note that if the patience is too small, underfitting might occur, and training may terminate too early due to stochastic fluctuations in the loss. Similarly, overfitting might likely occur when the patience is set to excessively high numbers (especially with a low number of simulations, since the loss function is typically more variable in this setting).

For the fine-tuning step of MF-NPE, no network weights were frozen. This choice has been purposely made to maintain full flexibility of the network to adapt to the high-fidelity model.

For the evaluation of MF-TSNPE-AF, we used 5 rounds of active sampling, where $80\%$ of the high-fidelity dataset was used for standard MF-NPE training, and $20\%$ was split across the rounds of active sampling. The active samples were selected using the acquisition function over an ensemble of 5 networks.

For a fair performance comparison, all methods were trained on the same datasets and evaluated on the same observations $x_o$. All amortized results were obtained over 10 network initializations, and all non-amortized results over 1 or 10 network initializations (depending on the computational cost of the task). We evaluated the methods over 30 observations for the C2ST metric, more than the 10 observations chosen previously for benchmarking (Lueckmann et al., 2021). This choice is motivated by our focus on evaluating the methods in low-data regimes, where greater certainty is required. The performance on the L5PC neuron task was evaluated with the metric NLTP and over 100 $x_o$'s. Here, the performance of the amortized methods was averaged over 10 network initializations, and in the non-amortized methods over 1 network initialization, since training had to be performed for each individual $x_o$. The performance of the methods on the recurrent spiking network task was averaged over 10 network initializations and evaluated over 262,008 observations, which was the maximum number of available samples for this high-dimensional problem.

---

[1] https://github.com/probabilists/zuko
[2] https://github.com/sbi-dev/sbi

# D TASKS

## D.1 OU PROCESS

The Ornstein-Uhlenbeck (OU) process is a high-fidelity model with 2 to 4 free parameters that contains a temporal structure in the observations. As a low-fidelity model, we chose i.i.d. samples from a Gaussian distribution (unstructured vector), parametrized by the mean and standard deviation. This setting makes it well-suited to examine the impact of parameter space overlap between the low- and high-fidelity models, as well as the impact of a systematic bias in the posterior of the low-fidelity model on transfer learning.

**High-fidelity model** The Ornstein-Uhlenbeck process models a drift-diffusion process of a particle starting at position X(0) and drifting towards an equilibrium state. The model has two main components: a *drift* term and a *diffusion* term:

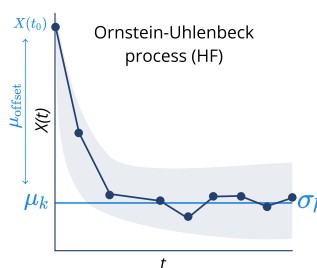

$$dX_t = \underbrace{\gamma(\mu - X_t)dt}_{\text{drift}} + \underbrace{\sigma dW_t}_{\text{diffusion}},$$

where $\mu$ is the mean of the asymptotic distribution over positions X, $\sigma$ is the magnitude of the stochasticity of the process and $\gamma$ is the convergence speed. $X(0)$ is the initial position of the process, which we assume to be stochastic: $X(0) \sim \mathcal{N}(\mu + \mu_{\text{offset}}, 1)$. The parameters of interest that we aim to estimate are $\mu, \sigma, \gamma, \mu_{\text{offset}}$.

The Ornstein-Uhlenbeck process was approximated with the Euler-Maruyama method:

$$X(t + \delta t) = X(t) + f_{\text{drift}}(t, X)\,\delta t + f_{\text{diffusion}}(t, X)\,\sqrt{\delta t}\,\mathcal{N}(0, 1).$$

Figure 5: The four parameters of the Ornstein-Uhlenbeck process: the mean $\mu$, standard deviation $\sigma$, convergence rate $\gamma$, and $\mu_{\text{offset}}$, which is the difference between the initial condition $X(0)$ and mean $\mu$.

Starting from the exact likelihood for the Ornstein-Uhlenbeck process given by Kou et al. (2012):

$$f_{\text{exact hi}}(\boldsymbol{X} \mid \mu, \gamma, \sigma) = \prod_{t=1}^{n} \frac{1}{\sqrt{\pi g}\sigma} \exp\left\{-\frac{1}{g\sigma^2}\left((\mu - X_t) - \sqrt{1 - \gamma g}\,(\mu - X_{t-1})\right)^2\right\},$$

where $g = (1 - \exp(-2\gamma\Delta t))/\gamma$, we modify it by incorporating an additional parameter $\mu_{\text{offset}}$ to account for a stochastic $X(0)$.

The full likelihood $f_{\text{exact hi}}(\boldsymbol{X} \mid \mu, \sigma, \gamma, \mu_{\text{offset}})$ is given by

$$f_{\text{exact hi}}(\boldsymbol{X} \mid \mu, \sigma, \gamma, \mu_{\text{offset}}) = \frac{1}{\sqrt{2\pi}} \exp\left\{-\frac{(x - (\mu + \mu_{\text{offset}}))^2}{2}\right\} f_{\text{exact hi}}(\boldsymbol{X} \mid \mu, \gamma, \sigma)$$

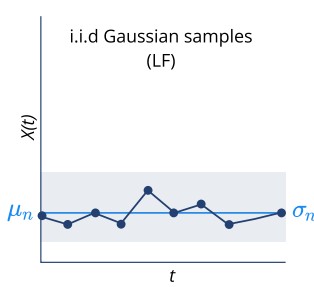

**Low-fidelity model** As a low-fidelity model, we use i.i.d. Gaussian Samples. At convergence, the distribution over $X_t$ approaches a Gaussian distribution with mean $\mu$ and standard deviation $\frac{\sigma}{\sqrt{2\gamma}}$. In our setup, we chose a low-fidelity model that corresponds to time-independent random draws from a Gaussian distribution with mean $\mu_{\text{lo}}$ and standard deviation $\sigma_{\text{lo}}$:

$$X_t \sim \mathcal{N}(\mu_{\text{lo}}, \sigma_{\text{lo}}^2) \qquad (6)$$

The posterior distribution over the parameters of the low-fidelity model has a biased mean influenced by the initial position $\mu_{\text{offset}}$ and convergence speed $\gamma$.

Figure 6: i.i.d. Gaussian samples with mean $\mu_{\text{L}}$ and standard deviation $\sigma_{\text{L}}$.

| | |
|---|---|
| **Prior** | $\mu \sim \mathcal{U}(0.1, 3), \quad \sigma \sim \mathcal{U}(0.1, 0.6), \quad \gamma \sim \mathcal{U}(0.1, 1), \quad \mu_{\text{offset}} \sim \mathcal{U}(0, 4)$ |
| **HF Simulator** | $\mathbf{x}|\theta = (x_1, \ldots, x_{101}), \quad x_0 \sim \mathcal{N}(\mu + \mu_{\text{offset}}, 1),$ where |
| | $dx_t = \gamma(\mu - x_t)dt + \sigma dW_t$ |
| **LF Simulator** | $\mathbf{x}|\theta = (x_1, \ldots, x_{10}), \quad x_i \sim \mathcal{N}(\mu_{\text{lo}}, \sigma_{\text{lo}}^2),$ |
| **HF Dimensionality** | $\theta \in \mathbb{R}^{2-4}, \quad \mathbf{x} \in \mathbb{R}^{101}, \quad U(\mathbf{x}) \in \mathbb{R}^{10}$ |
| **LF Dimensionality** | $\theta \in \mathbb{R}^{2}, \quad \mathbf{x} \in \mathbb{R}^{10}, \quad U(\mathbf{x}) \in \mathbb{R}^{10}$ |
| **References** | (Holý & Tomanová, 2022; Carter & Strey, 2023; Kou et al., 2012) |

For the two-dimensional experiment, the free parameters $\gamma, \mu_{\text{offset}}$ have been fixed to $\gamma = 0.5$ and $\mu_{\text{offset}} = 3.0$. For the three-dimensional-experiment, only $\mu_{\text{offset}} = 3.0$. The **summary statistics** $U(x)$ from the high-fidelity model consists of 10 uniformly distributed subsamples drawn from a trace of 101 timesteps. Parameters and summary statistics are illustrated in Figures 5 and 6.

### D.2 POSTERIOR DISTRIBUTIONS OVER OU PROCESS

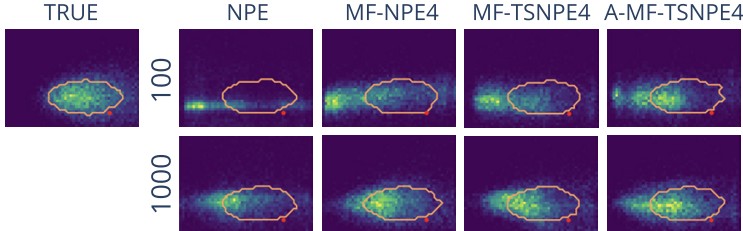

Figure 7: Posterior density estimates for a single observation from the OU process with two free parameters (OU2). The orange contour lines contain 68% of the probability mass of the true posterior distribution.

### D.3 OU PROCESS WITH VARYING PARAMETER SPACE

We present a comparison of our multifidelity approaches to NPE and MF-ABC, with different numbers of pre-trained low-fidelity simulations. MF-NPE3 is pre-trained on a low-fidelity dataset of size $10^3$, while MF-NPE4 and MF-NPE5 use datasets of $10^4$ and $10^5$ low-fidelity simulations, respectively. The MF-ABC results suggest that neural density approaches scale better to complex problems (Frazier et al., 2024).

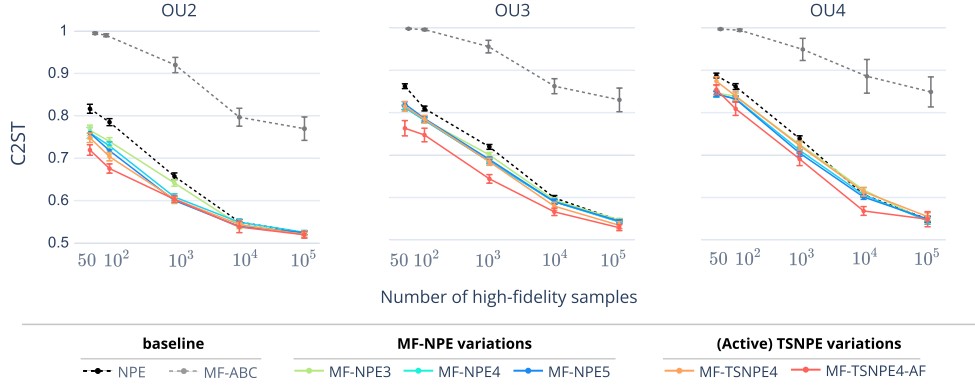

Figure 8: MF-NPE benefits from larger low-fidelity datasets. We ran MF-ABC with hyperparameters $\epsilon = (1, 1)$ and $\eta = (0.9, 0.3)$ (more details in Appendix E.1.1). All variants of our method perform better than MF-ABC and NPE.

## D.4 INFERRING THE PARAMETERS OF A GAUSSIAN MODEL PRETRAINED ON THE OU3 MODEL

In this example, we examine how the performance changes when the low-fidelity model has a larger number of parameters than the high-fidelity model: the low-fidelity model is the Ornstein-Uhlenbeck process with three parameters, and the high-fidelity model corresponds to i.i.d. Gaussian samples parameterised by a mean and variance (so, only two parameters). To accomplish that, the density estimator pre-trained on the low-fidelity model was fine-tuned only on the dimensions of the high-fidelity and the extra dimension was kept as a dummy dimension. NPE was directly trained on the 2-dimensional parameter space of the high-fidelity model. At inference time, the posterior evaluation was performed only on the high-fidelity parameter dimensions. We observe that when the dimension of $\theta$ is smaller than the dimension of $\theta_L$, transfer learning provides a significant improvement in performance.

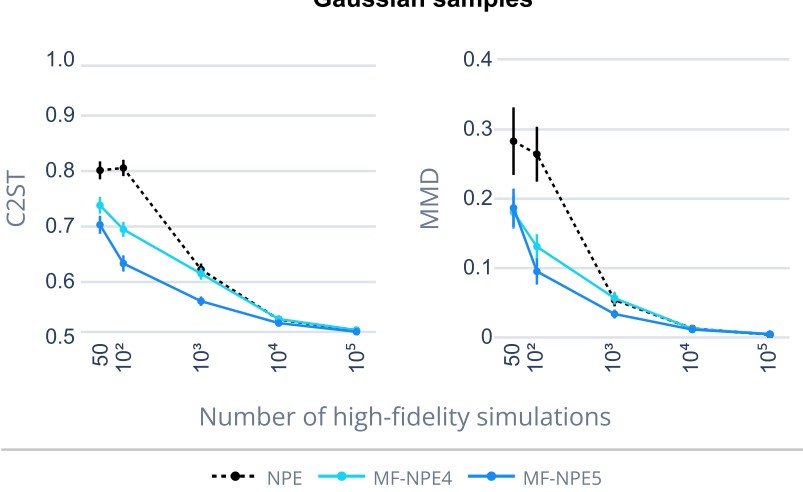

Figure 9: Evaluation with C2ST and MMD over a two-dimensional Gaussian Samples model, pretrained on the three-dimensional OU process model.

### D.5 SLCP

Simple Likelihood Complex Posterior (SLCP) is a benchmark inference task that has been artificially designed to have a simple likelihood, but a very non-trivial 5-dimensional posterior to infer. In this example, we study the impact of multifidelity in cases where the dimensionality of the parameter space differs between the low-fidelity and high-fidelity models.

**High-fidelity model**  The SLCP problem involves five parameters. The prior distribution is uniform across a five-dimensional parameter space, and the observations consist of four two-dimensional samples drawn from a Gaussian distribution. Both the mean and the variance of this Gaussian depend on the parameters through nonlinear mappings. The high-fidelity model follows the code in the SBI benchmarking paper (Lueckmann et al., 2021).

**Low-fidelity model**  In the low-fidelity model, we experimented with the effect of different numbers of parameters on the inference quality. We fixed $m_\theta = 0$, and kept the parameters of $S_\theta$ free.

| | |
|---|---|
| **Prior** | $\mathcal{U}(-3, 3)$ |
| **HF Simulator** | $\mathbf{x}\|\theta = (x_1, \dots, x_4), \quad x_i \sim \mathcal{N}(\mathbf{m}_\theta, \mathbf{S}_\theta),$ |
| | where $\mathbf{m}_\theta = \begin{bmatrix} \theta_1 \\ \theta_2 \end{bmatrix}, \quad \mathbf{S}_\theta = \begin{bmatrix} s_1^2 & \rho s_1 s_2 \\ \rho s_1 s_2 & s_2^2 \end{bmatrix},$ |
| | with $s_1 = \theta_3^2,\ s_2 = \theta_4^2,\ \rho = \tanh(\theta_5).$ |
| **LF Simulator** | $\mathbf{x}\|\theta = (x_1, \dots, x_4), \quad x_i \sim \mathcal{N}(0, \mathbf{S}_\theta),$ |
| | where $\mathbf{S}_\theta = \begin{bmatrix} s_1^2 & \rho s_1 s_2 \\ \rho s_1 s_2 & s_2^2 \end{bmatrix},$ |
| | with $s_1 = \theta_3^2,\ s_2 = \theta_4^2,\ \rho = \tanh(\theta_5).$ |
| **HF Dimensionality** | $\theta \in \mathbb{R}^5, \quad \mathbf{x} \in \mathbb{R}^8$ |
| **LF Dimensionality** | $\theta \in \mathbb{R}^3, \quad \mathbf{x} \in \mathbb{R}^8$ |
| **References** | (Papamakarios et al., 2019; Hermans et al., 2020) |
| | (Durkan et al., 2020; Greenberg et al., 2019; Lueckmann et al., 2021) |
| | (Thiele et al., 2025) |

### D.6 SIR

The Susceptible, Infected, and Recovered (SIR) model is a classical epidemiological benchmark example that captures the spread of infectious diseases through three interacting compartments: Susceptible (S), Infectious (I), and Recovered (R). Its dynamics are governed by the system of ordinary differential equations. The model is parameterized by two rates: the **infection rate** $\beta$ and the **recovery rate** $\gamma$. We investigate how multifidelity addresses the partly observed dynamics of the model. Rather than observing the three dynamics of the SIR model (following the setup of the SBI benchmarking (Lueckmann et al., 2021), we assume that no dynamics regarding the recovered subjects are known (SI model).

**Low-fidelity model**    In the low-fidelity model, we assume no information is available about the dynamics of recovered individuals. The total population size and the initial conditions are kept consistent with the high-fidelity model.

**Bounded domain**    $[0.001, 3]^2$

**Prior**    $\beta \sim \text{LogNormal}(\log(0.4), 0.5), \quad \gamma \sim \text{LogNormal}(\log(0.125), 0.2)$

**HF Simulator**    $\mathbf{x}|\theta = (x_1, \ldots, x_{50}), \quad x_i = I_i/N$ equally spaced,

$I$ is simulated from $\dfrac{dS}{dt} = -\beta\dfrac{SI}{N}, \quad \dfrac{dI}{dt} = \beta\dfrac{SI}{N} - \gamma I, \quad \dfrac{dR}{dt} = \gamma I$

**LF Simulator**    $\mathbf{x}|\theta = (x_1, \ldots, x_{50}), \quad x_i = I_i/N$ equally spaced,

$I$ is simulated from $\dfrac{dS}{dt} = -\beta\dfrac{SI}{N}, \quad \dfrac{dI}{dt} = \beta\dfrac{SI}{N} - \gamma I,$

**Dimensionality**    $\theta \in \mathbb{R}^2, \ x \in \mathbb{R}^{3 \times 161}, \ U(\mathbf{x}) \in \mathbb{R}^{10}$

**Fixed parameters**    Population size $N = 10^6$, duration of task $T = 160$ days.
Initial conditions: $(S(0), I(0), R(0)) = (N-1, 1, 0)$

**References**    (Lueckmann et al., 2021; Greenberg et al., 2019)
(Hermans et al., 2020; Durkan et al., 2020)

Summary statistics $U(x)$ are 10 subsamples from the $I$ trace.

## D.7 IMAGE EXAMPLE

We apply our method to a problem with **high-dimensional observations**, and explore the benefits of transfer learning in combination with **embedding networks**. The high-fidelity model is a 256x256 image, while the low-fidelity model has a resolution of 32x32. An example of both simulator outputs is shown in Fig. 10.

**High-fidelity model** The Gaussian Blob image example contains high-dimensional observations that have been embedded with a CNN embedding from the SBI package (Boelts et al., 2024). The model renders a 2D image, which we modeled as a 256 x 256 pixel image of a Gaussian blob, and aiming to infer three parameters ($\mu_{\text{off}}, \sigma_{\text{off}}, \gamma$): the horizontal and vertical displacements of the blob, and its contrast (Lueckmann et al., 2019). The image is in grey-scale and is generated through a binomial distribution with a total count of 255 and probability $p_{ij}$, as described in Lueckmann et al. (2019).

**Low-fidelity model** In our setup, the low-fidelity model generates a spatially low-resolution dataset (32x32 image). We upscale these images using interpolation techniques and provide the resulting low-resolution inputs to the embedding network $U(x)$.

| | |
|---|---|
| **Prior HF** | $x_{\text{off}}, y_{\text{off}} \sim \mathcal{U}(0, 256), \quad \gamma \sim \mathcal{U}(0.2, 2)$ |
| **Prior LF** | $x_{\text{off}}, y_{\text{off}} \sim \mathcal{U}(0, 32), \quad \gamma \sim \mathcal{U}(0.2, 2)$ |

**Simulator**
$$\mathbf{x}|\theta = (x_1, \ldots, x_{1024}), \quad \text{where,}$$
$$I_{xy} \sim \text{Bin}(\cdot|255, p_{xy})$$
$$p_{xy} = 0.9 - 0.8 \exp^{-0.5(r_{xy}/\sigma^2)^\gamma}$$
$$r_{xy} = (x - x_{\text{off}})^2 + (y - y_{\text{off}})^2$$

**Dimensionality HF** $\quad \theta \in \mathbb{R}^3, \ x \in \mathbb{R}^{256 \times 256}, U(x) \in \mathbb{R}^{32}$

**Dimensionality LF** $\quad \theta \in \mathbb{R}^3, \ x \in \mathbb{R}^{32 \times 32}, U(x) \in \mathbb{R}^{32}$

**Fixed parameters** Standard deviation $\sigma_{\text{lf}} = 2, \sigma_{\text{hf}} = 12$

**References** (Lueckmann et al., 2019)

# E    GAUSSIAN BLOB EVALUATION

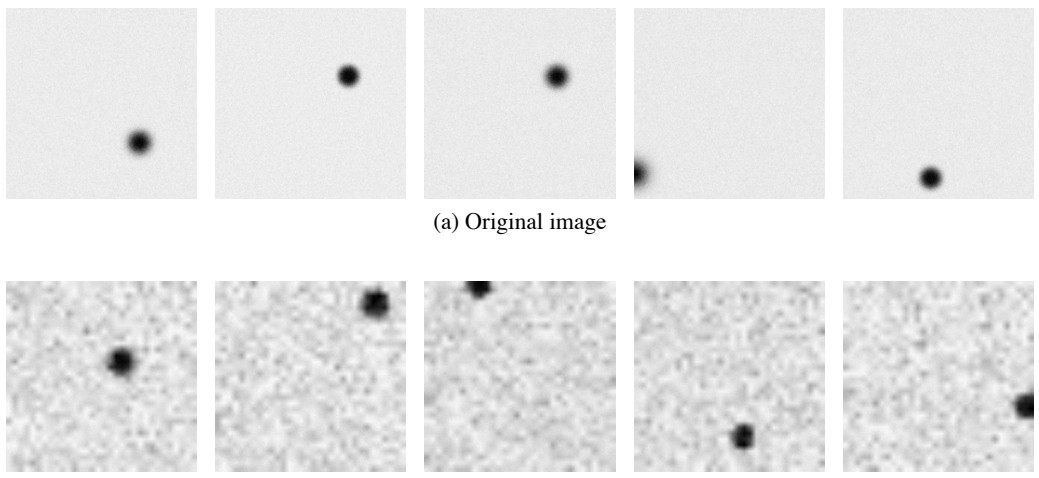

(a) Original image

(b) low-fidelity simulations

Figure 10: Five examples of generated images with the Gaussian Blob across the two fidelities, with (a) the original 256x256 high-fidelity simulations, (b) the upsampled 32x32 low-fidelity simulations.

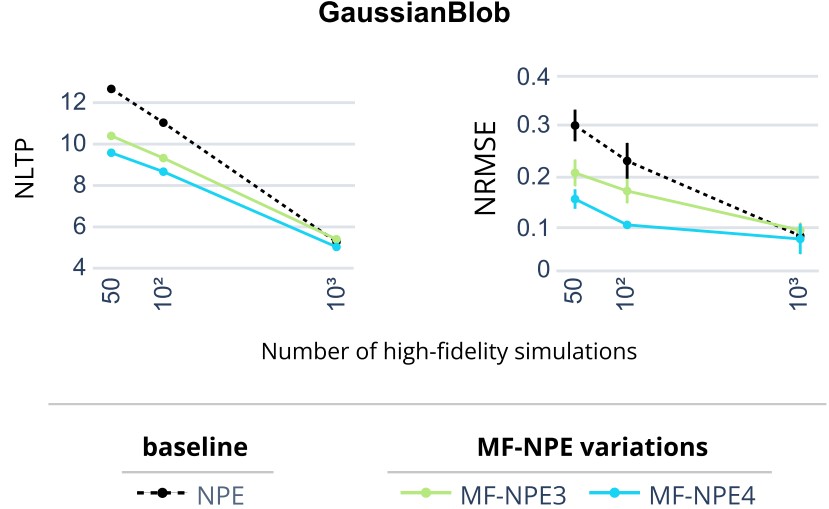

Figure 11: Method comparison with NLTP and NRMSE for the Gaussian Blob task. Evaluated over 10000 observations.

### E.1 Data generation and transformations for increased network performance

During the performance evaluation, we encountered numerical instabilities, particularly with NPE in low-simulation budgets: a substantial proportion of the estimated probability density was placed outside of the uniform prior bounds, a phenomenon dubbed 'leakage' that has been previously documented (Greenberg et al., 2019; Deistler et al., 2022). Logit-transforming the model parameters before training the density estimator resolved the issue.

This transformation creates a mapping from a bounded to an unbounded space, resulting in a density estimation within the prior bounds after the inverse transformation. In addition, the summary statistics of the simulations were z-scored for improved performance of the density estimator, the default setting in the SBI package (Boelts et al., 2024).

### E.1.1 MULTIFIDELITY APPROXIMATE BAYESIAN COMPUTATION (MF-ABC)

We translated into Python a publicly available Julia implementation of the multifidelity ABC algorithm (Prescott & Baker, 2020). In our setup, the adaptive sampling scheme of MF-ABC selected approximately $30\%$ of the batch size as high-fidelity samples in the OU2 and OU3 tasks, and $50\%$ in the OU4 task. To ensure consistency with our neural network experiments, we z-scored the simulator output before inference. We also explored the effect of varying the acceptance threshold $\epsilon$. We found that the hyperparameters slightly affect the performance of MF-ABC, but that MF-NPE always shows superior performance than MF-ABC (Figure 12). However, MF-ABC has several other hyperparameters to tune. We cannot exclude the hypothesis that larger performance gains could be obtained from such an approach by a more extensive hyperparameter search.

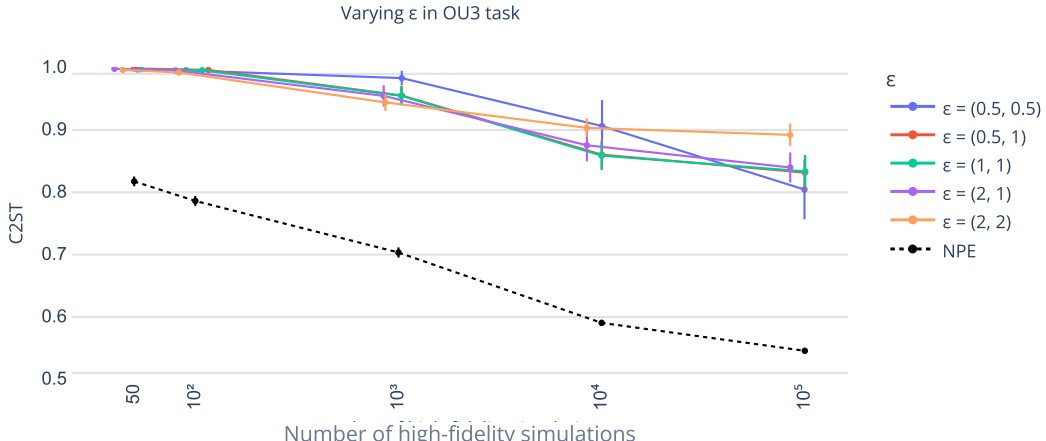

Figure 12: **C2ST results for MF-ABC with varying hyperparameters $\epsilon$.** Mean and $95\%$ confidence interval.

**MF-ABC posteriors** ABC-based methods typically require a significantly larger number of samples for convergence (Lueckmann et al., 2021; Frazier et al., 2024). In line with previous studies, we find that $10^4$ samples are not yet enough for MF-ABC to converge to a good estimate of the posterior in the OU2 task.

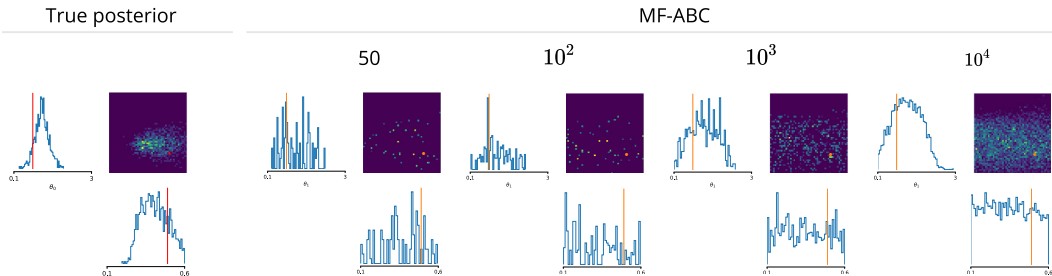

Figure 13: **Comparison between MF-ABC posterior estimates and the true posterior.** Results for the Ornstein-Uhlenbeck process with two free parameters. Posterior estimates are shown for varying numbers of high-fidelity simulations (50, 100, $10^3$, and $10^4$).

## F    TASK 2: MULTICOMPARTMENTAL SINGLE NEURON MODEL

The response of a morphologically detailed neuron to an input current is typically modeled with a multicompartmental neuron model wherein the voltage dynamics of each compartment $\mu$ are based on Hodgkin-Huxley equations (Hodgkin & Huxley, 1952):

$$
\begin{aligned}
c_{\mathrm{m}} \frac{dV_\mu}{dt} = & -i_{\mathrm{m}}^\mu + \frac{I_{\mathrm{e}}^\mu}{A_\mu} + g_{\mu,\mu+1}\left(V_{\mu+1} - V_\mu\right) \\
& + g_{\mu,\mu-1}\left(V_{\mu-1} - V_\mu\right).
\end{aligned}
\tag{7}
$$

The total membrane current $i_{\mathrm{m}}$ for a specific compartment is the sum over different types of ion channels $i$, such as sodium, potassium and leakage channels:

$$
i_{\mathrm{m}} = \bar{g}_{\mathrm{Na}}m^3h(V - E_{\mathrm{Na}}) + \bar{g}_{\mathrm{K}}n^4(V - E_{\mathrm{K}}) + \bar{g}_{\mathrm{L}}(V - E_{\mathrm{L}}) + \bar{g}_{\mathrm{M}}p(V - E_{\mathrm{M}})
\tag{8}
$$

We are interested in inferring the densities of two prominent ion channels $\bar{g}_{\mathrm{Na}}$ and $\bar{g}_{\mathrm{K}}$.

The low- and high-fidelity models differ in the number of compartments per branch: the low-fidelity model has a single compartment per branch, while the high-fidelity model consists of eight compartments per branch.

All simulations were performed using Jaxley (V 0.8.2) (Deistler et al., 2025) over 120 ms. The injection current is a step current of 0.55mV over 100 ms, with a delay of 10ms. The step size of the simulator is 0.025.

When sampling from the prior distribution over parameters, approximately $0.05 - 0.1\%$ of the respective simulations had clearly unrealistic summary statistics: these simulations were iteratively replaced by random draws from the prior distribution/proposal or active learning list (depending on the algorithm) until we collected a desired number of valid simulations.

### F.1 NRMSE EVALUATION

In addition to the NLTP metric, we demonstrate that the NRMSE metric yields results that support our conclusions.

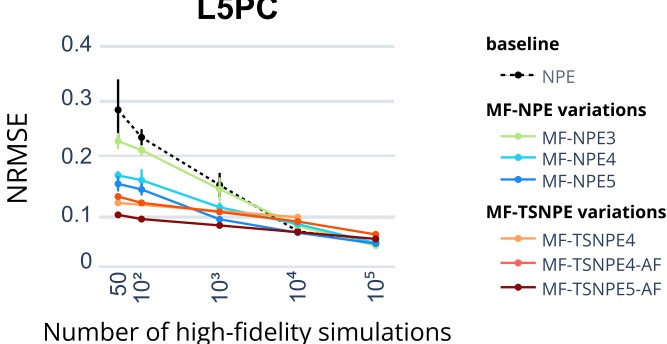

Figure 14: NRMSE evaluation for the multicompartmental neuron model.

### F.2 SIMULATION-BASED CALIBRATION AND POSTERIOR DISTRIBUTIONS

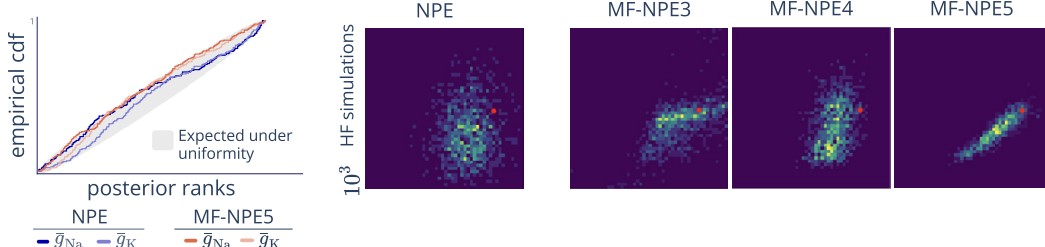

Figure 15: Simulation-based calibration (left) and respective posterior distributions for NPE and MF-NPE (right) for the multicompartmental neuron model task. MF-NPE is respectively, pretrained on $10^3, 10^4, 10^5$ low-fidelity simulations (dubbed as MF-NPE3, MF-NPE4, and MF-NPE5). All models were trained on $10^3$ high-fidelity simulations.

### F.3 Posterior Predictive Checks

With only 50 high-fidelity simulations, MF-NPE gives similar accuracy to NPE trained on 1000 simulations (Fig. 16), and for a fixed number of 1000 high-fidelity simulations, MF-NPE5 outperforms NPE (Fig. 17).

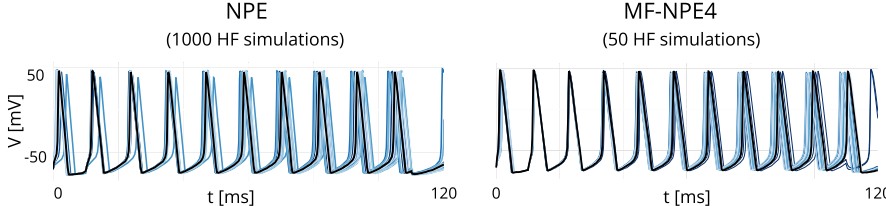

Figure 16: Posterior predictives for the multicompartmental neuron model with varying number of high-fidelity simulations.

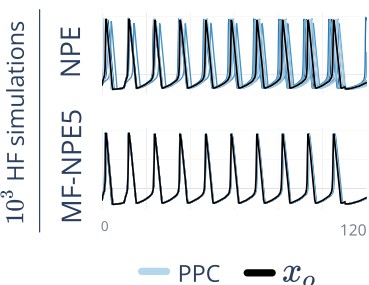

Figure 17: Posterior predictives for the multicompartmental neuron model for a fixed number of high-fidelity simulations.

### F.4 Low and high-fidelity traces

We present simulations with the models with 1- and 8-compartments per dendritic branch (low- and high-fidelity models, respectively) to illustrate that the model outputs are different, given the same parameters.

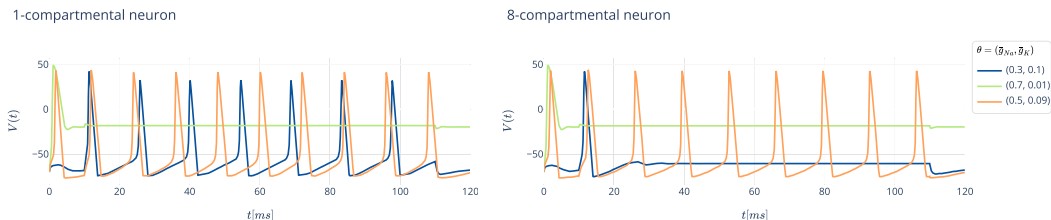

Figure 18: Simulated membrane potential traces of an L5 pyramidal cell (L5PC) model with Jaxley (Deistler et al., 2025). The low- and high-fidelity models are, respectively, a single-compartment model per dendritic branch versus an eight-compartment model per branch.

## G    TASK 3: SPIKING NETWORK MODEL

### G.1    HIGH-FIDELITY MODEL

We considered a recurrent spiking network of 5120 neurons (4096 excitatory, 1024 inhibitory), with parameters taken from Confavreux et al. (2023). The membrane potential dynamics of neuron $j$, excitatory ($E$) or inhibitory ($I$), followed

$$\tau_m \frac{dV_j}{dt} = -(V_j - V_{\text{rest}}) - g_j^{\text{E}}(t)(V_j - E_{\text{E}}) - g_j^{\text{I}}(t)(V_j - E_{\text{I}}), \tag{9}$$

A postsynaptic spike was generated whenever the membrane potential $V_j(t)$ crossed a threshold $V_j^{\text{th}}(t)$, with an instantaneous reset to $V_{\text{reset}}$. This threshold $V_j^{\text{th}}(t)$ was incremented by $V_{\text{spike}}^{\text{th}}$ every time neuron $j$ spiked and otherwise decayed following

$$\tau_{\text{th}} \frac{dV_j^{\text{th}}}{dt} = V_{\text{base}}^{\text{th}} - V_j^{\text{th}}. \tag{10}$$

The excitatory and inhibitory conductances, $g^{\text{E}}$ and $g^{\text{I}}$ evolved such that

$$g_j^{\text{E}}(t) = a g_j^{\text{AMPA}}(t) + (1-a) g_j^{\text{NMDA}}(t) \quad \text{and} \quad \frac{dg_j^{\text{I}}}{dt} = -\frac{g_j^{\text{I}}}{\tau_{\text{GABA}}} + \sum_{i \in \text{Inh}} w_{ij}(t) \delta_i(t)$$

$$\text{with} \quad \frac{dg_j^{\text{AMPA}}}{dt} = -\frac{g_j^{\text{AMPA}}}{\tau_{\text{AMPA}}} + \sum_{i \in \text{Exc}} w_{ij}(t) \delta_i(t) \quad \text{and} \quad \frac{dg_j^{\text{NMDA}}}{dt} = \frac{g_j^{\text{AMPA}}(t) - g_j^{\text{NMDA}}}{\tau_{\text{NMDA}}}, \tag{11}$$

with $w_{ij}(t)$ the connection strength between neurons $i$ and $j$ (unitless), $\delta_k(t) = \sum \delta(t - t_k^*)$ the spike train of pre-synaptic neuron $k$, where $t_k^*$ denotes the spike times of neuron $k$, and $\delta$ the Dirac delta. All neurons received input from 5k Poisson neurons, with 5% random connectivity and constant rate $r_{\text{ext}} = 10$Hz in each simulation. The recurrent connectivity was instantiated with random sparse connectivity (10%). All recurrent synapses in the network ($E$-to-$E$ and $E$-to-$I$, $I$-to-$E$, $I$-to-$I$) underwent variations of spike-timing dependent plasticity (STDP) (Gerstner & Kistler, 2002; Confavreux et al., 2023). Given the learning rate $\eta$, the weights between the neurons $i$ and $j$ of connection type $X$-to-$Y$ evolved over time as:

$$\frac{dw_{ij}}{dt} = \eta \left[ \delta_{\text{pre}}(t)(\alpha + \kappa x_{\text{post}}(t)) \right.$$
$$\left. + \delta_{\text{post}}(t)(\beta + \gamma x_{\text{pre}}(t)) \right]. \tag{12}$$

with variables $x_i(t)$ and $x_j(t)$ describing the pre- and postsynaptic spikes over time:

$$\frac{dx_i}{dt} = -\frac{x_i}{\tau_{\text{XY}}^{\text{pre}}} + \delta_i(t) \quad \text{and} \quad \frac{dx_j}{dt} = -\frac{x_j}{\tau_{\text{XY}}^{\text{post}}} + \delta_j(t) \tag{13}$$

with $\tau_{\text{XY}}^{\text{pre}}$ and $\tau_{\text{XY}}^{\text{post}}$ the time constants of the traces associated with the pre- and postsynaptic neurons, respectively.

The 24 free parameters of interest were $\tau_{\text{pre}}, \tau_{\text{post}}, \alpha, \beta, \kappa, \gamma$ multiplied by the number of synapse types (e.g., $\alpha_{EE}, \alpha_{II}, \alpha_{EI}, \alpha_{IE}$), following previous work (Confavreux et al., 2023).

### G.2    LOW-FIDELITY MODEL

Following previous work (Confavreux et al., 2023; Vogels et al., 2011; Dayan & Abbott, 2001), a (partial) mean-field theory applied to the $E$-to-$E$ and $E$-to-$I$ connections in the model described above gave:

$$r_{\text{E}}^* = \frac{-\alpha_{\text{EE}} - \beta_{\text{EE}}}{\lambda_{\text{EE}}} \quad \text{and} \quad r_{\text{I}}^* = \frac{-\alpha_{\text{EI}} r_{\text{E}}^*}{\beta_{\text{EI}} + \lambda_{\text{EI}} r_{\text{E}}^*} \tag{14}$$

with $r_{\text{E}}^*$ and $r_{\text{I}}^*$ the firing rates of the excitatory (resp. inhibitory) population at steady state, and

$$\lambda_{\text{XY}} = \kappa_{\text{XY}} \tau_{\text{XY}}^{\text{post}} + \gamma_{\text{XY}}^{\text{pre}} \tag{15}$$

With type $(X, Y) \in \{E, I\}$. For all synapse types, we assume $(-\alpha_{\text{XY}} - \beta_{\text{XY}}) > 0$ and $\lambda_{\text{XY}} > 0$, as a second-order stability condition (Confavreux et al., 2023). Note that in this low-fidelity model, we only considered 2 of the 4 plastic conditions, and thus 12 of the 24 free parameters of the high-fidelity model.

### G.3 SYNAPTIC PLASTICITY WITH VARYING PARAMETER SPACE

We investigated how inference performance changes as the discrepancy between the low- and high-fidelity models increases. To this end, we varied the dimensionality of the low-fidelity model between 3, 6, and 12 parameters, while keeping the high-fidelity model fixed at 24 parameters. Parameters that were excluded from inference in the low-fidelity settings were fixed to the following values for each connection type: $\tau_{\mathrm{pre}} = \tau_{\mathrm{post}} = 0.05, \gamma = -1.9, \alpha = \beta = \kappa = 0.5$. The value of $\gamma$ should be smaller than other parameters to fulfill the second-order stability condition (Confavreux et al., 2023).

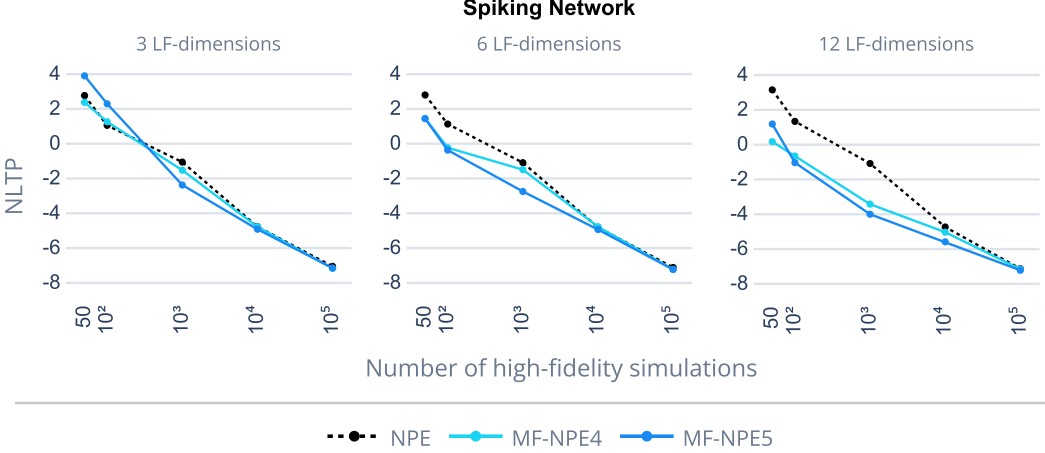

Figure 19: Negative-log-likelihood over true parameters, with different numbers of free parameters in the low-fidelity model.

We observe that the performance of MF-NPE degrades as the number of parameters in the low-fidelity model decreases as compared to the high-fidelity model. In particular, unlike in all our other experiments, when the low-fidelity model had only 3 parameters, pretraining on $10^5$ low-fidelity samples led to worse MF-NPE performance: in this regime, using $10^5$ samples (MF-NPE5) resulted in negative transfer, whereas pretraining on $10^4$ samples (MF-NPE4) resulted in a performance close to standard NPE.

### G.4    DISCUSSION ON ALTERNATIVE SOLUTIONS

We consider the following strategies:

- pretraining on solely low-fidelity simulations,
- pretraining on the joint of low- and high-fidelity simulations.

### G.4.1    PRETRAINING ON LOW FIDELITY SAMPLES

This approach follows the main discussion in Sec. 3.1.1, and has also been the main method employed in the paper. We purposefully do not freeze the weights after transfer, allowing the network to retain the flexibility to adapt to high-fidelity simulations.

### G.4.2    PRETRAINING ON THE JOINT OF LF AND HF SAMPLES

We examined whether pretraining on the joint distribution of low- and high-fidelity simulations could provide a better initialization for subsequent fine tuning. As shown in Fig. 20, this strategy yields no significant improvement on the first two benchmarking tasks compared to standard MF-NPE. However, we encourage further work to investigate additional variations on this approach to improve the domain adaptation (e.g., domain adaptation through MMD Elsemüller et al. (2025), importance weighting for extremely unbalanced datasets, adversarial discriminative domain adaptation, training a single multifidelity inference network).

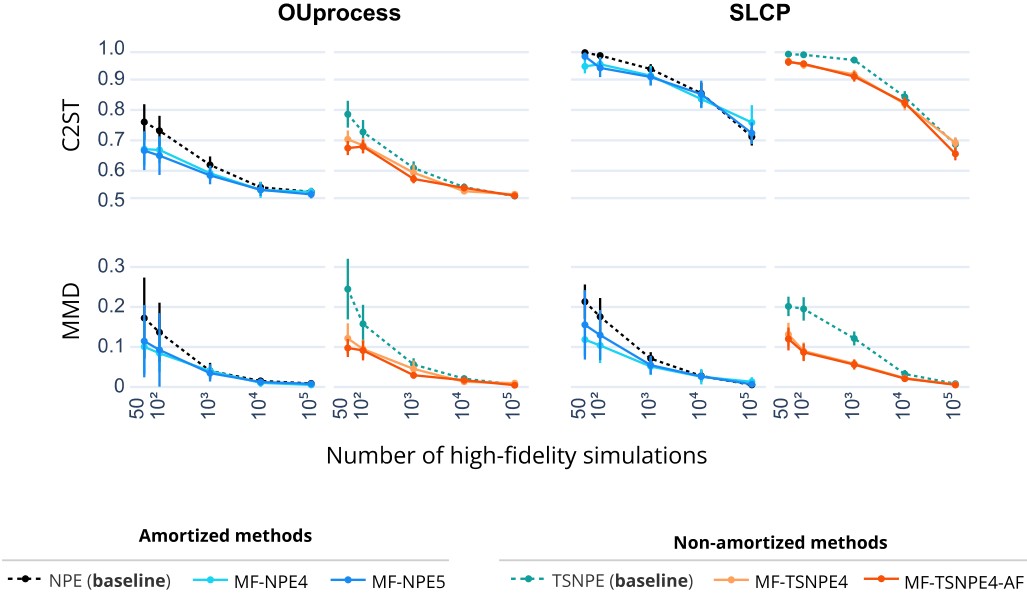

Figure 20: MF-(TS)NPE (joint) has been pretrained on both low- and high-fidelity samples.

## H  PRIOR BOUNDS ACROSS NEUROSCIENCE TASKS

For the OU process task, we chose a uniform prior with bounds that would lead to a range of different outputs. For the multicompartment neuron model task, we chose a uniform prior with bounds based on the work of Deistler et al. (2022). For the spiking network model task, we chose a uniform prior with bounds based on the work of Confavreux et al. (2023).

Table 1: Prior bounds for the single- and multicompartmental neuron model.

| PARAMETER NAME | LOWER BOUND | UPPER BOUND |
|:---:|:---:|:---:|
| $\bar{g}_{\mathrm{NA}}$ | 0.005 | 0.8 |
| $\bar{g}_{\mathrm{K}}$ | $10^{-6}$ | 0.15 |

Table 2: Prior bounds for each synapse type ($E$-to-$E$, $E$-to-$I$, $I$-to-$E$ and $I$-to-$I$) for the spiking neural network and mean-field model.

| PARAMETER NAME | LOWER BOUND | UPPER BOUND |
|:---:|:---:|:---:|
| $\tau_{\mathrm{pre}}$ | 0.01 | 0.1 |
| $\tau_{\mathrm{post}}$ | 0.01 | 0.1 |
| $\alpha$ | $-2$ | 2 |
| $\beta$ | $-2$ | 2 |
| $\gamma$ | $-2$ | 2 |
| $\kappa$ | $-2$ | 2 |

# I DISTANCE BETWEEN THE LF AND HF POSTERIOR

Both the low and high-fidelity posterior distributions have been trained on $10^5$ simulations and evaluated over 10 true observations. In the table below, we focus on cases with two fidelities and measure the distance between the low and high-fidelity models with the MMD and C2ST metrics. We observe that the distance between the posterior distributions is not a direct measure of success in transfer learning. For instance, the posterior distributions of the low-and high-fidelity models of the L5PC neuron are significantly different. However, the network still manages to leverage information between the two simulators (Figure 3), supporting the theoretical results of Tahir et al. (2024).

Transfer learning seems to work less well on the OU process task when the dimensionality of the parameters differs between the low- and high-fidelity models (see Sec. 8). This is observed despite the fact that the distance between the low and high-fidelity posteriors is lower for the OU4 case than for the OU2 case, as the low-fidelity OU2 posterior is highly biased (Fig. 21).

Table 3: Distance between low- and high-fidelity posterior (mean $\pm$ std) for different tasks.

| Task | MMD | C2ST |
|---|---|---|
| SLCP | $0.13 \pm 0.05$ | $0.91 \pm 0.03$ |
| SIR | $0.04 \pm 0.03$ | $0.57 \pm 0.03$ |
| OU2 | $1.00 \pm 0.11$ | $0.98 \pm 0.02$ |
| OU3 | $0.69 \pm 0.087$ | $0.98 \pm 0.01$ |
| OU4 | $0.24 \pm 0.05$ | $0.90 \pm 0.04$ |
| L5PC | $0.76 \pm 0.23$ | $0.99 \pm 0.00$ |
| SynapticPlasticity | $0.01 \pm 0.00$ | $0.70 \pm 0.02$ |

## I.1 PAIRPLOTS

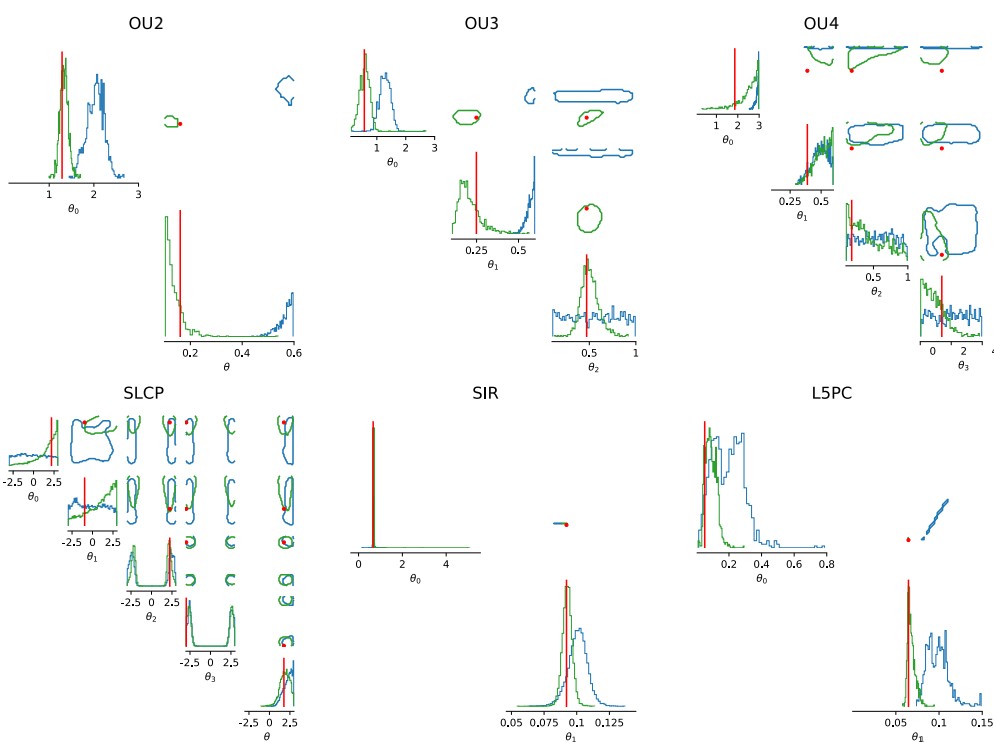

Figure 21: Posterior distributions of the low-fidelity posterior (blue) and high-fidelity posterior (green). Contours contain 68% of the true posterior mass for the low-fidelity model. Vertical bars and dots correspond to the value of the true parameters.

## J    SIMULATION VERSUS TRAINING COST

We tracked the wall-clock run-time for training and simulation stages of the neural density estimator. Computations were performed on nodes each equipped with 4× Intel Xeon Gold 6448H CPUs (32 cores per socket, 128 physical cores, 256 logical CPUs) and approximately 2TB RAM, running Linux 5.14.0. We compare the costs in regimes where the performance of NPE is similar to MF-NPE and MF-TSNPE-AF (Fig. 3). Details about the network architecture and hyperparameters are described in Appendix C.1. In cases where many samples had to be generated for active non-amortized schemes (e.g., $10^5$ HF samples for the L5PC task; Figure 3), we used multiprocessing over CPUs. The simulations for the third task were parallelized over 913 CPUs.

Table 4: Comparison of methods for the real-world tasks in terms of the number of simulations and computational cost. Total training cost is reported as mean ± standard deviation over 5 network runs.

|         | method      | # simulations | | CPU (seconds) | | |
|---------|-------------|-----|-----|----------------|-------------------|-------------------|
|         |             | LF  | HF  | tot. cost (sim.) | tot. cost (train) | total cost |
| L5PC    | NPE         | NA  | $10^4$ | 4940 | 70.39 ± 18.32 | 5010.39 ± 18.32 |
|         | MF-NPE      | $10^4$ | $10^3$ | 1032 | 96.94 ± 15.19 | 1128.94 ± 15.19 |
|         | MF-TSNPE-AF | $10^4$ | 50 | 607 | 557.44 ± 52.5 | 1164.44 ± 52.5 |
| Network | NPE         | NA  | $10^4$ | $3 \times 10^6$ | 120.43 | 3,000,120 |
|         | MF-NPE      | $10^5$ | $10^3$ | $3 \times 10^5$ | 94.54 | 300,094 |

Table 5: Comparison of methods across models in terms of the number of simulations and accuracy. Evaluated using the NLTP metric.

|         | Method      | # Simulations | | Accuracy (C2ST/NLTP) |
|---------|-------------|-----|-----|-----------------------|
|         |             | LF  | HF  |                       |
|         | NPE         | NA  | $10^4$ | -5.87 ± 0.04 |
|         | MF-NPE      | $10^4$ | $10^3$ | -5.73 ± 0.05 |
| L5PC    | MF-TSNPE-AF | $10^4$ | 50 | -5.08 ± 0.27 |
| Network | NPE         | NA  | $10^4$ | -4.72 ± 0.01 |
|         | MF-NPE      | $10^5$ | $10^3$ | -4.08 ± 0.01 |

Table 4 shows that the multifidelity approaches make sense when the training cost is significantly lower than the simulation cost, such as in the L5PC and the spiking network model. For instance, in the spiking network task, a single high-fidelity simulation requires approximately 5 CPU minutes, whereas a low-fidelity simulation takes only 0.0008 seconds.

# K    TARP EVALUATION FOR ALL TASKS

We performed additional evaluations on the calibration of all experiments with TARP (Lemos et al., 2023).

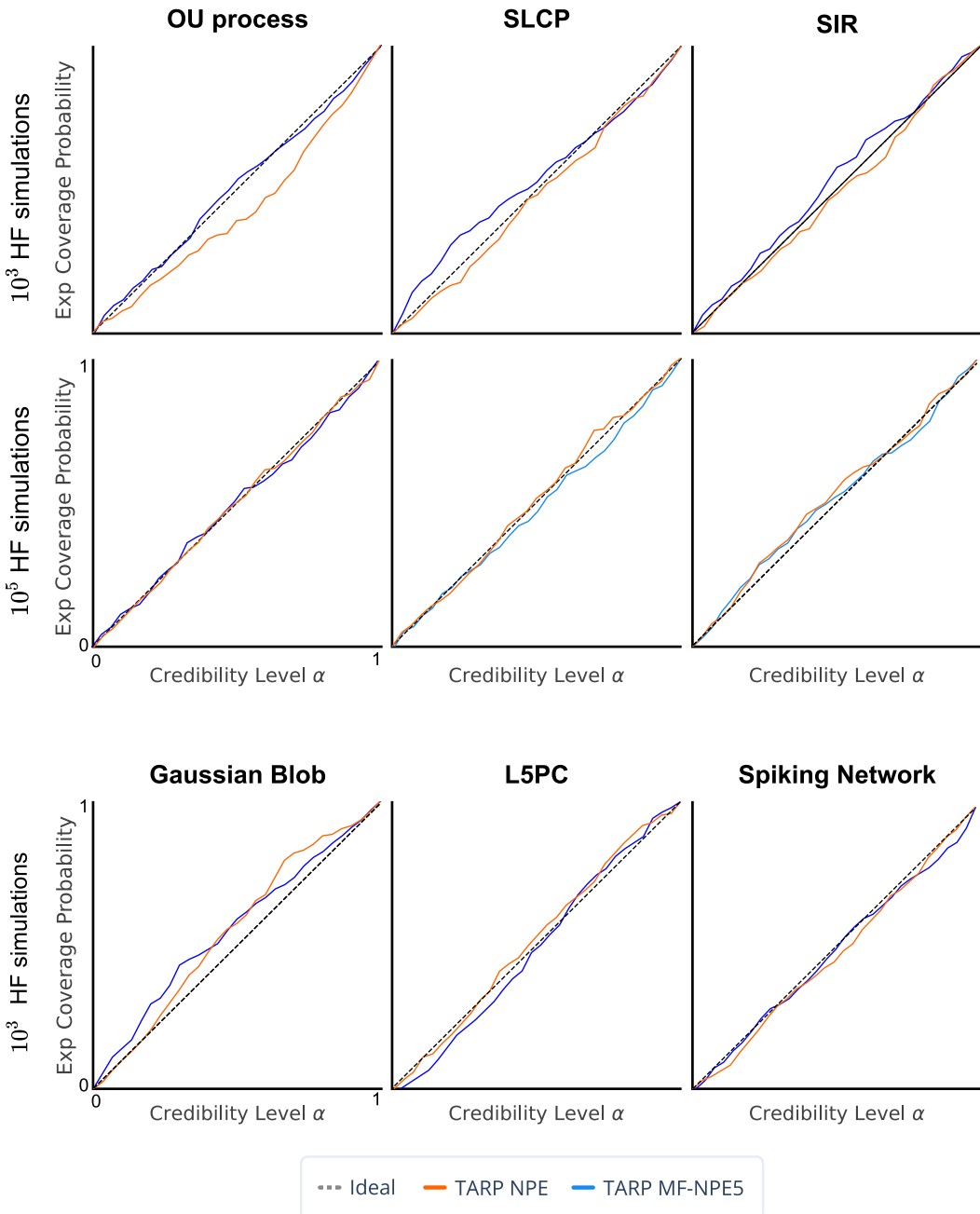

Figure 22: TARP calibration test across $10^5$ LF simulations ($10^4$ for the Gaussian blob example). The calibration test was performed over 200 runs.

## L  MF-NPE FOR MULTIPLE LOWER-FIDELITY SIMULATORS

---

**Algorithm 2** MF-NPE with multiple fidelities

---

1: **Input:** Simulations $\{(\boldsymbol{\theta}, \boldsymbol{x}^{(f)})\}_{f=1}^{F}$ over $F$ fidelities; Early stopping criterion $S$; conditional density estimators $\{q_\psi^{(f)}(\boldsymbol{\theta}|\boldsymbol{x}^{(f)})\}_{f=1}^{F}$ with features $\psi$.

2: **for** $f = 1$ **to** $F$ **do**

3: $\quad \mathcal{L}(\psi^{(f)}) = \frac{1}{N^{(f)}} \sum_{i=1}^{N^{(f)}} -\log q_\psi^{(f)}\left(\boldsymbol{\theta}_i|\boldsymbol{x}_i^{(f)}\right).$

4: $\quad \text{opt}^{(f)} \leftarrow \text{Adam}(\cdot)$

5: $\quad$ **if** $f > 1$ **then**

6: $\quad\quad$ Initialize $q_\psi^{(f)}$ with features of trained $q_\psi^{(f-1)}$.

7: $\quad$ **end if**

8: $\quad$ **for** epoch in epochs **do**

9: $\quad\quad$ train $q_\psi^{(f)}$ to minimize $\mathcal{L}(\psi^{(f)})$ until $S$ is reached.

10: $\quad$ **end for**

11: **end for**

---

# M  SEQUENTIAL ALGORITHMS

## M.1  MF-TSNPE

---

**Algorithm 3** MF-TSNPE

---

1: **Input:** $N$ pairs of $(\boldsymbol{\theta}, \boldsymbol{x}_{\mathrm{L}})$; conditional density estimators $q_\psi(\boldsymbol{\theta}|\boldsymbol{x}_{\mathrm{L}})$ and $q_\phi(\boldsymbol{\theta}|\boldsymbol{x})$ with learnable parameters $\psi$ and $\phi$; early stopping criterion $S$; simulator $p(\boldsymbol{x}|\boldsymbol{\theta})$; prior $p(\boldsymbol{\theta})$; number of rounds $R$; $\epsilon$ that defines the highest-probability region (HPR$_\epsilon$); number of high-fidelity simulations per round $M$.
2: **Output:** posterior estimate $q_\phi(\boldsymbol{\theta}|\boldsymbol{x})$
3: $\mathcal{L}(\psi) = \frac{1}{N} \sum_{i=1}^{N} -\log q_\psi\left(\boldsymbol{\theta}_i|\boldsymbol{x}_i^{\mathrm{L}}\right)$ .
4: **for** epoch in epochs **do**
5:     train $q_\psi$ to minimize $\mathcal{L}(\psi)$ until $S$ is reached.
6: **end for**
7: Initialize $\tilde{p}(\boldsymbol{\theta})$ as $p(\boldsymbol{\theta})$
8: Initialize $q_\phi$ with weights and biases of trained $q_\psi$.
9: **for** r in $R$ **do**
10:     $\boldsymbol{\theta}^{(r)} \sim \tilde{p}(\boldsymbol{\theta})$, sample parameters from proposal
11:     $\boldsymbol{x}^{(r)} \sim p(\boldsymbol{x}|\boldsymbol{\theta}^{(r)})$, generate high-fidelity simulations
12:     **for** epoch in epochs **do**
13:         $\mathcal{L}(\phi) = \frac{1}{M} \sum_{i=1}^{M} -\log q_\phi\left(\boldsymbol{\theta}_i^{(r)}|\boldsymbol{x}_i^{(r)}\right).$
14:         train $q_\phi$ to minimize $\mathcal{L}(\phi)$ until $S$ is reached.
15:     **end for**
16:     Compute expected coverage $(\tilde{p}(\boldsymbol{\theta}), q_\phi)$
17:     $\tilde{p}(\boldsymbol{\theta}) \propto p(\boldsymbol{\theta}) \cdot \mathbb{1}_{\theta \in \mathrm{HPR}_\epsilon}$
18: **end for**

---

All experiments were run with $R = 5$ rounds and $\epsilon = 1e^{-6}$. More details about TSNPE at Deistler et al. (2022).

## M.2 MF-TSNPE-AF

---

**Algorithm 4** MF-TSNPE-AF

---

1: **Input:** $N$ pairs of $(\boldsymbol{\theta}, \boldsymbol{x}_{\mathrm{L}})$; conditional density estimator $q_\psi(\boldsymbol{\theta}|\boldsymbol{x}_{\mathrm{L}})$ with learnable parameters $\psi$ and ensemble of conditional density estimators $\{q_\phi^e(\boldsymbol{\theta}|\boldsymbol{x})\}_E^e$, each with independent $\phi$; early stopping criterion $S$; simulator $p(\boldsymbol{x}|\boldsymbol{\theta})$; prior $p(\boldsymbol{\theta})$; number of rounds $R$; $\epsilon$ that defines the highest-probability region (HPR$_\epsilon$); number of high-fidelity simulations per round $M$.

2: **Output:** Ensemble posterior estimate $q_\phi(\boldsymbol{\theta}|\boldsymbol{x}) = \frac{1}{E}\sum_{e=1}^E q_\phi^e(\boldsymbol{\theta}|\boldsymbol{x})$

3: $\mathcal{L}(\psi) = \frac{1}{N}\sum_{i=1}^N -\log q_\psi\left(\boldsymbol{\theta}_i|\boldsymbol{x}_i^{\mathrm{L}}\right)$ .

4: **for** epoch in epochs **do**

5:     train $q_\psi$ to minimize $\mathcal{L}(\psi)$ until $S$ is reached.

6: **end for**

7: **for** e $\in$ Ensemble **do**

8:     Initialize $q_\phi^e$ with weights and biases of trained $q_\psi$.

9: **end for**

10: $\boldsymbol{\theta}_{\mathrm{pool}} \sim p(\boldsymbol{\theta})$

11: Initialize $\tilde{p}(\boldsymbol{\theta})$ as $p(\boldsymbol{\theta})$

12: **for** r in $R$ **do**

13:     $\boldsymbol{\theta}_{\mathrm{prop}}^{(r)} \sim \tilde{p}(\boldsymbol{\theta})$, generate $M - B$ samples from proposal

14:     $\boldsymbol{\theta}_{\mathrm{active}}^{(r)} =$ top $B$ values from $\boldsymbol{\theta}_{\mathrm{pool}}$ using the acquisition function equation 2

15:     $\boldsymbol{\theta}^{(r)} = \{\boldsymbol{\theta}_{\mathrm{prop}}^{(r)} \cup \boldsymbol{\theta}_{\mathrm{active}}^{(r)}\}$

16:     $\boldsymbol{x}^{(r)} \sim p(\boldsymbol{x}|\boldsymbol{\theta}^{(r)})$, generate high-fidelity simulations

17:     **for** e $\in$ Ensemble **do**

18:       **for** epoch in epochs **do**

19:         $\mathcal{L}(\phi) = \frac{1}{M}\sum_{i=1}^M -\log q_\phi^e\left(\boldsymbol{\theta}_i^{(r)}|\boldsymbol{x}_i^{(r)}\right)$.

20:         train $q_\phi$ to minimize $\mathcal{L}(\phi)$ until $S$ is reached.

21:       **end for**

22:     **end for**

23:     Compute expected coverage $(\tilde{p}(\boldsymbol{\theta}), \frac{1}{E}\sum q_\phi^e(\boldsymbol{\theta}|\boldsymbol{x}))$

24:     $\tilde{p}(\boldsymbol{\theta}) \propto p(\boldsymbol{\theta}) \cdot \mathbb{1}_{\theta \in \mathrm{HPR}_\epsilon}$

25: **end for**

---

All experiments were run with $R = 5$ rounds, $\epsilon = 1e^{-6}$, and an ensemble of 5 networks. The addition of an acquisition function biases the proposal distribution, causing the density estimate to diverge from the true posterior. In principle, this could be addressed by using atomic proposals (Greenberg et al., 2019), but given that such an approach suffers from posterior leakage, we do not introduce a proposal correction in order to retain the well-behaved loss function in TSNPE. We argue that the benefit of informative samples would outweigh the potential bias, as long as the percentage of samples selected from the acquisition function would be small compared to the proposal samples. Therefore, we set $B = .2M$ to mitigate the concern of biasing the posterior with parameters selected with the acquisition function.

