# OpenReview forum: "Multifidelity Simulation-based Inference for Computationally Expensive Simulators"
_ICLR.cc/2026/Conference — ICLR 2026 Poster_

### Official Review · Reviewer_Sge5 · 2025-10-28

**Soundness:** 3
**Presentation:** 4
**Contribution:** 4
**Rating:** 8
**Confidence:** 4

**Summary:**

The paper proposes MF-(TS)NPE, a multi-fidelity recipe for SBI that pre-trains a neural posterior estimator on cheap low-fidelity simulations and fine-tunes it using a small budget of high-fidelity simulations. Then they propose two enhanced variants:  a sequential variant that further performs sequential SBI starting from the fine-tuned model and using the high-fidelity simulator and an active variant (A-MF-TSNPE) that adds an acquisition rule based on the variance of an ensemble of fine-tuned models.
Across SBI benchmarks (SIR, SLCP, OU, Gaussian Blob) and two computational neuroscience tasks (multicompartment neuron; large recurrent spiking network), the method matches or improves accuracy while using orders of magnitude fewer HF simulations; wall-clock accounting shows substantial real compute savings when HF simulations dominate cost (Table 4). The paper also investigates when transfer helps via controlled OU perturbations, highlighting the roles of mutual information and representational coherence between fidelities (Fig. 4).

**Strengths:**

-**Clarity and motivation**: The paper is clearly written and well motivated: leveraging multiple simulators with varying levels of fidelity seems like a natural way to improve efficiency of SBI. While it was considered in the ABC literature, it received little attention when it comes to deep networks for SBI.

-**Simplicity**: The proposed approach based on fine-tuning is quite simple to implement and seems to yield effective improvements. This simplicity makes it broadly applicable accross tasks, as shown in the exeperiments, and easily adaptable to other settings: sequential and active settings.

-**Soundness**: The paper proposes an empirical characterization for when such approaches should be effective based on mutual information and feature alignment. This supports the soundness of the approach although precise theoretical characterization remains unclear, but could be the subject of future work.

**Weaknesses:**

The limitations of this work are well discussed in section 5.
**Reliance on LF-HF alignment**: The method relies on a-priori alignment between the low-fidelity and high-fidelity simulators, but in general it is hard to quantify such alignment. While the paper studies the effect of this alignment, it is done on a toy example/controlled setting. Currently, the approach relies on domain experts that can know beforehand whether simulators are aligned, but it would be could to have a procedure or some guidelines for non-experts. Perhaps a better theoretical understanding could achieve that.

**Questions:**

Is there a way to know in advance (without evaluation on real data), if the low-fidelity simulator can be helpful?

---

> ### Author Response · Authors · 2025-11-21
>
> We sincerely thank Reviewer Sge5 for the thoughtful feedback and for the positive assessment of the paper’s motivation, clarity, and contributions. We are pleased that the reviewer found the proposed multifidelity approach broadly applicable and empirically well supported.
>
> # W1. On the reliance on low–high fidelity alignment.
> We appreciate the reviewer’s point that our method benefits from an underlying alignment between the low- and high-fidelity simulators, and that formalizing or quantifying this alignment is challenging. While we agree that a theoretical characterization would be valuable, doing so requires addressing several unresolved theoretical questions in neural posterior estimation and transfer learning. As the reviewer notes, such work is nontrivial and lies outside the scope of this submission, which focuses on establishing a practical and flexible multifidelity method with rigorous empirical validation.
>
> Importantly, our empirical analyses (Sec. 4.4; Fig. 4) provide concrete insights into what kinds of alignment matter in practice. By perturbing the OU process in controlled ways, we identify two factors that govern when transfer meaningfully helps:
>
> - **Mutual information** between low- and high-fidelity observations.
> - **Representational coherence**, i.e., similarity in how simulators encode task-relevant structure.
>
> These experiments show (i) that transfer succeeds when both conditions hold, (ii) that mutual information alone is not sufficient (e.g., invertible distortions can preserve information but harm transfer), and (iii) that these factors have clear empirical signatures detectable from observed simulations (with no recourse to observed data). While not a complete theory, this analysis provides insight to guide practitioners.
>
> # Q1. Is there a way to know in advance if the low-fidelity simulator can be helpful?
> A universal metric is unlikely, because the usefulness of pretraining depends on task-specific structure. However, our experiments suggest several practical heuristics that provide some insight.
>
> Our empirical results indicate that transfer learning is effective when the following conditions hold:
> - There is substantial overlap in the parameter spaces of the simulators; the low- and high-fidelity models share a common subset of parameters. This pattern appears consistently in the empirical results, particularly in Fig. 7 and Fig. 19.
>
> - High mutual information between simulated observations. In the OU experiments, transfer effectiveness degrades smoothly as we artificially reduce information shared between fidelities. Mutual information can be estimated from simulated data alone (e.g., via MINE), providing a practical diagnostic.
>
> - Coherent feature representations. Even if information is preserved, representational distortions (e.g., dimension reversal in Fig. 4) significantly harm transfer. Inspecting embeddings or summary statistics for structural alignment can help identify good candidates for low-fidelity models.
>
> We again thank the reviewer for the thoughtful and constructive feedback. Their comments helped us clarify key points and strengthen the presentation, and we appreciate their contribution to improving the manuscript.

---

> > ### Comment · Reviewer_Sge5 · 2025-11-26
> >
> > Thank you for your response. I have read the other reviews, and while I agree with the points made by reviewer Eu4Y, that a  theoretical analysis would be valuable, I also think that a well-carried proof of concept, even empirical, as done here, is equally valuable and can motivate further theoretical investigations.
> > I maintain my score

---

### Official Review · Reviewer_CW6v · 2025-10-30

**Soundness:** 3
**Presentation:** 4
**Contribution:** 4
**Rating:** 8
**Confidence:** 5

**Summary:**

This paper introduces MF-(TS)NPE, a novel and efficient approach to simulation-based inference (SBI) designed to tackle the computational burden of parameter estimation in expensive high-fidelity simulators. The method leverages multifidelity modeling – combining computationally cheap, low-fidelity simulations with a limited number of high-fidelity simulations – and transfer learning to accelerate Bayesian inference. An active learning extension, A-MF-TSNPE, further improves efficiency by strategically selecting high-fidelity evaluations. Demonstrated across benchmark tasks and challenging neuroscience applications, MF-(TS)NPE achieves up to two orders of magnitude reduction in required high-fidelity simulations while maintaining comparable accuracy to existing methods like NPE and TSNPE. The work provides a valuable tool for efficient Bayesian analysis in a wide range of scientific domains reliant on complex simulations.

**Strengths:**

This paper demonstrates a strong commitment to rigorous and accessible research in simulation-based inference. A key strength lies in its exceptionally clear and well-motivated introduction to both the challenges of SBI and the foundational techniques like TSNPE, significantly enhancing the paper’s accessibility to a broad audience. The experimental design is particularly commendable, building logically from established benchmarks to increasingly complex, real-world simulations (including L5PC and spiking neural networks), providing robust validation of the proposed method.

The authors showcase a thorough analysis, supported by a diverse suite of metrics – even extending to appendix materials to accommodate the breadth of results – and consistently present findings with appropriate uncertainty estimates, empowering readers to critically evaluate the conclusions. Further contributing to clarity, the writing is commendably free of unnecessary jargon and complex simulators are described in sufficient detail to facilitate a basic level of reproducibility. Finally, the authors adopt a refreshingly honest and nuanced approach, presenting their work as a solid incremental improvement within the field rather than an unsubstantiated claim of state-of-the-art performance. This measured presentation strengthens the credibility and impact of the research.

**Weaknesses:**

This paper, while strong overall, exhibits a few areas for improvement. The introduction occasionally suffers from verbosity, dedicating excessive space to establishing the context of simulation-based inference – particularly in the opening paragraphs – which could be streamlined for greater impact. More critically, the authors sometimes present conclusions that exceed the support provided by the empirical results, necessitating more cautious interpretation. Finally, the quantification of improvements, specifically regarding computational cost, sometimes relies on relative gains rather than absolute metrics, hindering a full appreciation of the practical benefits achieved by the proposed method. Addressing these issues would enhance the clarity, rigor, and impact of the presented work.

**Questions:**

- lines 30-53: nice introduction to the field, could be cut shorter perhaps
- lines 200-202: "Overall, MF-NPE shows competitive performance compared ..." this sentence confused me as no strong results in favor of this claim was presented up to here to a reader, perhaps reconsider the formulation or remove this passage
- page 4, Algorithm 1: it might be helpful to clearly identify low-fidelity and high-fidelity related parts of the algorithm (perhaps hints to the right of the algorithm)
- page 4, Algorithm 1: line numbers would be helpful to communicate with peers about this
- line 240-241: clearly state that you are interested in the epistemic uncertainty (makes the text more approachable to people experienced in UQ)
- line 246-248: "Alternatively, one could ... Griesmer 2024)". MCdropout is a simple but often unstable UQ method, I personally would remove this sentence
- on section 3.1.3: comment briefly on the fact that any UQ method at use here needs to be battle proven in the OOD regime (this is a hard problem within its own right, hence only comment)
- line 268: "In the limit of a large number of pairs, ..." the high fidely simulation dataset comprises at minimum 50 samples. Would we expect convergence of NLTP to the KL here?
- line 295-300: I guess, you do not use TARP or methods alike because you are working in a low data regime?
- figure 2: do NOT include an interpretation in bold letters as the caption of the figure. Please leave it to the text (subsequently the reader) to make such assertions.
- line 320: "Across experiments, we observed a consistent performance increase with MF-NPE compared to NPE" This is not what I deduce from the figure. From all 12 cases under study, by eye, I can tell that in only 4 MF-NPE is below the NPE (black dashed line) with some distance larger than the error bars. Please adapt this statement, if my interpretation is correct.
- line 326: I can hardly compare Appendix F.1.1 to e.g. Fig 2. Please create a plot where this is possible and put it in F.1.1
- line 362: "approx. 4 times" please be precise; also provide an absolute scale for reference on what 4 times higher means
- line 373: perhaps add a second model critique methdo like TARP etc
- section 4.3: all citations lack hyperrefs in my PDF reader, please check

---

> ### Author Response · Authors · 2025-11-21
>
> We thank the reviewer for their clear, detailed, and constructive feedback. We appreciate that the reviewer found the work novel, clearly written and well-motivated. We discuss the raised points below and updated the manuscript accordingly.
>
> # Q1 and W1: Reducing verbosity and focusing on SBI.
> We shortened the introductory section about Bayesian inference and transitioned more quickly to SBI and neural posterior estimation. We now discuss NPE and TSNPE in greater detail than other methods to guide the attention of the reader towards the contributions of our work. We also refocused the transfer learning section to emphasize its relevance for numerical simulators rather than generic applications.
>
> # Q2-Q6. Clarifications in Section 3 and Algorithm 1
> Lines 200–202: this passage has been removed
> - Algorithm 1:
>   - We added labels clearly distinguishing low-fidelity and high-fidelity steps.
>   - We added line numbers to improve readability and allow easy referencing.
> - Lines 240–241: We now explicitly state that we target epistemic uncertainty.
> - Lines 246–248: We removed the sentence about MC dropout, as suggested.
>
> # Q7. UQ in OOD settings: comment briefly on the fact that any UQ method at use here needs to be battle proven in the OOD regime
> We kindly request the reviewer to please elaborate on this comment, as we are unsure we fully understand it.
>
> # Q8. Convergence of NLTP
> As suggested by the reviewer, NLTP can be shown to converge to the KL divergence between the true and the estimated posteriors (up to a constant) when averaged over an infinite number of true parameters and respective true observations (Lueckmann et al. 2021, https://arxiv.org/abs/2101.04653). Note that in our experiments, the number of evaluations that guarantees a good estimation of NLTP was substantially larger than the number of training samples for the neural density estimators (which was as low as 50 samples in some training regimes).
>
> # Q9. Use of TARP / calibration
> We added TARP-based calibration checks to all tasks and included the results in Appendix K. SBC plots for the L5PC task are in the Appendix.
>
> # Q10-Q11.  Figure 2 caption and performance claims
> In Figure 2, across the three benchmarking tasks, we see that MF-NPE4 (light blue) and MF-NP5 (dark blue) are always below NPE (black line), and similarly for the sequential methods, MF-TSNPE4 and MF-TSNPE4-AF are always below TSNPE (green dotted line). We observe a similar trend in the fourth benchmark of the Gaussian blob, which is described in Figure 10. The only exception is the MMD evaluation of the SIR model at $10^3$ high-fidelity simulations for MF-TSNPE4 and MF-TSNPE4-AF, although the error bars are very large.
>
> # Q12-15.
> - **Comparison plots**: We added the NPE evaluation to the plot in Appendix E.1.1 and scaled it similarly to all other plots (on a range of 0.5-1).
> - **Approximately 4 times**: We adjusted the text to the exact value (4.44 ± 0.06).
> - **Additional model critique (TARP)**: We added this critique for all tasks, see Appendix K.
> - **Missing hyperrefs**: We checked the PDF and confirmed that hyperlinks are present in common viewers (Acrobat, browser-based); if issues remain, please let us know.
>
> We thank the reviewer again for the insightful comments, which have helped improve the clarity, rigor, and presentation of the manuscript.

---

> > ### Comment · Reviewer_CW6v · 2025-11-27
> > **Re Q7: How to validate UQ on OOD without loss of generalisation**
> >
> > My apologies for responding so late. The other reviews have had their impact on my time budget.
> >
> > I wrote:
> >
> > > on section 3.1.3: comment briefly on the fact that any UQ method at use here needs to be battle proven in the OOD regime (this is a hard problem within its own right, hence only comment)
> >
> > What I mean by this is, that many people motivate their use of UQ methods by aspiring to detect OOD samples. In practice, this well motivated goal is very hard to achieve. The reason is that often (if not always) you train on IID samples and also validate the UQ on these (albeit on a validation set). However, by curse of trying to assess data that is OOD (hence is not guaranteed to lie on the training manifold), you cannot be sure whether your UQ method is well calibrated for all possible OOD samples or working at all in this regime.
> >
> > Many authors sample from well-motivated but constructed OOD manifolds/pdfs and check their confidence intervals here. But to me this only provides a sufficient guarantee that/if your UQ method is working to a trustworthy degree. From a combinatorial perspective, there are simply many more possible OOD configurations than there are IID configurations of the data (especially when constraints to the problem space are removed).
> >
> > I personally feel that this is a common challenge that many authors are aware of, but is not discussed too often. Hence, I recommend refining the text in this section in this regard - but don't feel obliged to do more.

---

> > > ### Author Response · Authors · 2025-12-01
> > >
> > > We thank the reviewer for clarifying this point. We fully agree that UQ methods validated on IID data cannot guarantee good behavior on arbitrary OOD samples. As the reviewer writes, our method does not aim to detect OOD data. The uncertainty is used only to guide high-fidelity simulation selection within the simulator’s domain. We have added a short clarification in the paper to avoid any misunderstanding.

---

### Official Review · Reviewer_G9kC · 2025-10-31

**Soundness:** 3
**Presentation:** 2
**Contribution:** 3
**Rating:** 6
**Confidence:** 4

**Summary:**

The paper takes a few simple existing techniques and applies them to SBI. Transfer learning, active learning, and (already often applied) sequential training. They are applied specifically to multi fidelity simulators, which are plausibly useful to practitioners. The paper shows benchmark results and application-specific results in neuroscience. There are also many empirical attempts to characterize the effects of their contributions which are confusingly put in the appendix.

**Strengths:**

# results
- Figure 3 looks compelling. It shows the typical result that active learning decrease NLTP and also reduces calibration. It would be nice to know if it's problematically uncalibrated, but maybe an expert in the area can understand that.
- lines 439-448 are also super interesting. Why is this not in the main text? There is ~1.5 pages of application specific info about neuroscience, but this ML-general information is left out. Please include general info in the main text.

**Weaknesses:**

# writing
- It would be helpful if you stated your contribution more clearly at the beginning. There is a ton of background that obfuscates what algorithmic pieces you are introducing. It seems like the idea mainly exists in lines 282-232 and 234 - 248. I think a summary of that info could come significantly earlier.
- "A-MF-TSNPE" is super confusing because A often means amortized in this taxonomy while S implies sequential (aka active learning). There is already "active learning" going on in TSNPE. Usually amortized methods do not use active learning.

# results
- In Figure 2 it seems like you add a lot of algorithmic complexity for small gains. Meanwhile Figure 8 seems super novel and interesting. Please consider including Figure 8 in the main text.
- Where is an example of different numbers of parameters in the high fidelity and lower fidelity settings? That could be made more clear in the text. I know it's in the appendix.
- In general I think you can spend much more space on the benchmark results rather than all the background.

# citations
- Since you compare to and extend TSNPE, I would strongly recommend citing TMNRE https://arxiv.org/abs/2107.01214
- Additionally there is a huge literature about multi-fidelity bayesian optimization. Maybe it's worth citing that as well?

**Questions:**

- What happens when your cheap simulator has a narrow posterior compared to your high-fidelity simulator? It seems you are implicitly assuming that the cheap posterior is absolutely continuous w.r.t. the high fidelity simulator, but this doesn't have to be the case.
- You mention that the method is "applicable in situations where the low-fidelity model has fewer parameters than the high-fidelity model." What about the other way around?
- Why do you spend so much main text time on background rather than getting into the method more quickly and showing all your cool results? So many interesting experiments are in the appendix.

---

> ### Author Response · Authors · 2025-11-21
>
> We thank the reviewer for their clear and constructive feedback. We appreciate that the reviewer found the section “When does pre-training help?” compelling. In response, we moved this material from the appendix to the main text (Sec. 4.4), highlighting its importance to the paper.
> We also examined the calibration of MF-TSNPE5-AF (according to TARP) and found that the algorithm is relatively well-calibrated (Figure 3C).
>
> **Writing and structure**: We have now summarized our main algorithmic contributions in the opening paragraph of the Methods section and reduced background material so that readers can reach the method and results more quickly. Following your suggestion, we also renamed A-MF-TSNPE to MF-TSNPE-AF (AF meaning Acquisition Function) to avoid confusion regarding “active” versus “amortized” terminology.
>
> **Results and presentation**: Following the reviewer's recommendation, we included Figure 4 alongside the discussion on “why does pretraining help?” in the main text and reduced the background section. We also stated more explicitly in the main text that the figures of the OU process with varying numbers of parameters are in Appendix D.3, and added an experiment with a varying numbers of parameters for the Spiking Network (Appendix G.3).
>
> **Citations**:  We now cite TMNRE in the introduction, alongside TSNPE. We also expanded the discussion of related multifidelity Bayesian optimization approaches and added corresponding citations in the background section.
>
> # Q1. What happens when your cheap simulator has a narrow posterior compared to your high-fidelity simulator? It seems you are implicitly assuming that the cheap posterior is absolutely continuous w.r.t. the high fidelity simulator, but this doesn't have to be the case.
> We do not assume that the low-fidelity posterior is broader or absolutely continuous with respect to the high-fidelity posterior. Importantly, because our method uses **transfer learning** (and not sampling-based reweighting), MF-NPE remains applicable even when the low-fidelity model is out-of-distribution relative to the high-fidelity one. The effectiveness of transfer learning depends more on the similarity of the learned feature representations than on that of the predicted posterior densities (https://arxiv.org/abs/2410.08194). This implies that, even if the low-fidelity posterior is narrower, it may still require similar features to perform posterior estimation.
> As shown in the OU experiments (pairplots in Fig..21), the low-fidelity posterior can be highly skewed and, in some cases, assign almost zero mass to the region of the high-fidelity posterior. Yet, despite this mismatch, initializing from low-fidelity training still yields substantial performance improvements (Fig. 2). This empirically demonstrates that useful representational structure can transfer even when the distributions differ markedly.
>
> # Q2. You mention that the method is "applicable in situations where the low-fidelity model has fewer parameters than the high-fidelity model." What about the other way around?
> Transfer learning continues to provide significant benefits, even more than in the inverse setting where the high-fidelity model has more parameters. These results are presented in Sec. “D.4 Inferring the parameters of a Gaussian model pretrained on the OU3 model”, where pretraining on a higher-dimensional OU model improves inference for a lower-dimensional Gaussian samples task.
>
> # Q3. Why do you spend so much main text time on background rather than getting into the method more quickly and showing all your cool results? So many interesting experiments are in the appendix.
> We substantially shortened the introduction and background, allowing us to bring the method forward and move several important empirical results and discussions (including the transfer learning analysis) into the main text.
>
> We thank the reviewer again for their valuable feedback, which allowed us to further improve the validation of our methods.

---

> ### Comment · Reviewer_G9kC · 2025-11-25
> **reframing of the structure**
>
> Thank you for making all those changes.
>
> I see your point about the distribution's absolute continuity. I do not think you are doing reweighing... However, an extremely thorough analysis would disentangle these factors: feature learning versus having a good "starting place" for the density transformed by the normalizing flow.
> - If it is about feature learning, it reminds me of this paper about information bottleneck https://arxiv.org/pdf/1503.02406. How can we be sure this is the reason?
> - It also seems quite possible to me that having density in the "right place" to start with is also plausibly important. If density starts in "the right place," that implies that you effectively produce more good training data during sampling the expensive simulator. Meanwhile if there was a systematic bias in the parameters, you may produce mostly useless training data from the expensive simulator (after pretraining on the less expensive one).
>
> Anyhow, my points above are a bit complex and I don't really expect the authors to be able to fully answer them by the due date. I still feel positively about the paper and believe my score is fair.

---

> > ### Author Response · Authors · 2025-12-01
> >
> > Thank you for the positive and thoughtful engagement.
> >
> > # If it is about feature learning, it reminds me of this paper about information bottleneck https://arxiv.org/pdf/1503.02406. How can we be sure this is the reason?
> >
> > We agree that the information bottleneck provides an interesting framework to reason about the dynamics of transfer learning between fidelities. This question would involve studying how the feature representations change throughout the network and with transfer learning. As the reviewer writes, this would require an extensive, albeit relevant, analysis beyond the scope of the current work.
> >
> > # Density in the "right place" to start with could also be plausibly important for sampling
> >
> > In the amortized setting (MF-NPE), the initial state of the posterior distribution does not affect the generation of high-fidelity samples, since the samples are drawn from the same prior distribution.
> >
> > In standard SNPE, it is true that having very few samples in the initial round can cause the neural density estimator to converge to a local optimum where a substantial amount of posterior mass sits outside the true posterior, which then leads to suboptimal sampling in subsequent rounds. This is a known issue for all sequential sbi methods (e.g., SNPE), not just MF-TSNPE. The reviewer is right to point out that in our multifidelity sequential methods, this effect can also be influenced by the initialization from the low-fidelity density estimator. However, it is important to keep in mind that in our multifidelity sequential methods, the sampling in subsequent rounds is based on the estimated high-fidelity posterior, not the low-fidelity posterior. We should also note that in our numerical experiments, we did not find any cases where the sequential methods substantially underperformed compared to the amortised setting.

---

### Official Review · Reviewer_Eu4Y · 2025-11-01

**Soundness:** 3
**Presentation:** 3
**Contribution:** 3
**Rating:** 4
**Confidence:** 4

**Summary:**

The paper introduces an approach that uses transfer learning from cheap simulations to enable efficient posterior estimation for expensive higher-fidelity simulators. The method is evaluated on benchmark tasks and neuroscience applications, showing up to 100x reduction in required high-fidelity simulations. The basic idea is to pre-train a neural density estimator on cheap low-fidelity simulations, then fine-tunes on expensive high-fidelity simulations.

**Strengths:**

1. Addresses a real bottleneck in SBI where simulators can take minutes to hours per run. Relevant to lots of domains, in addition to those explored in the paper also in the physics sciences.
2. Exploration along the full SBI "stack" -- provides both amortized (MF-NPE) and sequential (MF-TSNPE) variants plus active learning (A-MF-TSNPE). Algorithms clearly specified.
3. Empirical results of when transfer works are strong -- beyond just demonstrating improvements, authors look at the underlying mechanisms (mutual information, representational coherence). The controlled experiments manipulating the OU process parameters provide insights into transfer learning limits.

**Weaknesses:**

1. https://arxiv.org/abs/2507.00514 and https://arxiv.org/abs/2505.21215 seem very similar in technique to the present paper (with a different application focus). While mentioned as concurrent work, this takes away from some of the methodological novelty somewhat.
2. Along the same lines, overall, a fairly straightforward, empirical application paper. Even toy theoretical analyses on transfer learning bounds -- perhaps relating low/high-fidelity similarity to sample complexity reduction -- would significantly strengthen the contribution. The empirical analysis in Section 4.4 is valuable but doesn't substitute for formal guarantees.
3. Possible minor notation inconsistencies, e.g. switches between p(x|\theta) and p_H(x|\theta)

**Questions:**

1. Table 4 shows A-MF-TSNPE training takes ~10x longer than MF-NPE -- when is this tradeoff worthwhile?
2. What are the major differentiators wrt https://arxiv.org/abs/2507.00514 and https://arxiv.org/abs/2505.21215 (e.g. the sequential, active learning parts)?
3. The spiking network has 12/24 parameters for low/high-fidelity, curious how performance degrades as this gap increases (guessing this is strongly domain dependent, but curious if there are general principles that could help guide applications)

---

> ### Author Response · Authors · 2025-11-21
>
> We thank the reviewer for the thoughtful and constructive feedback. We appreciate the comments on relevance, empirical strength, and coverage of multiple SBI variants. Below, we directly address the key concerns raised and highlight how we have improved the manuscript in response:
>
> # Q1. Table 4 shows A-MF-TSNPE training takes ~10x longer than MF-NPE -- when is this tradeoff worthwhile?
> This tradeoff can be worthwhile when the simulation time is much longer than the training time. In the L5PC setting, the high-fidelity model is only approximately 4 times more expensive than the low-fidelity model, and therefore A-MF-TSNPE does not yield computational gains over MF-NPE. In some neuroscience settings, it may be desirable to use neuron models with more compartments than chosen in this task, which could lead to substantial increases in computational cost, thereby naturally favouring the use of A-MF-TSNPE.
>
> # Q2 + W2. Concurrent work
> We appreciate the reviewer pointing to these concurrent works, which confirm that transfer learning for multifidelity SBI is a timely topic. We now more clearly articulate the differences:
>
> - **Scope and diversity of tasks**: In Saoulis (2025), the principle of transfer learning was evaluated on only one physics application (the hydrodynamic CAMELS multifield Dataset). Based solely on their results, it is unclear whether this approach would scale to other applications (e.g., handcrafted summary statistics, time series, etc.). By contrast, our work distinguishes itself by systematically investigating transfer learning and various task properties that might impact its performance, using different metrics.
> - **Analysis of when transfer learning helps**: Both preprints lack a discussion on the effectiveness of transfer learning as a function of the similarity between simulators.
> - **Dimensionality-mismatch experiments**: None of the preprints conducted systematic dimensionality-mismatch experiments. We demonstrate how transfer learning can degrade in a low-dimensional example (OU process) and a high-dimensional example (Spiking Network), where in the first example, the number of parameters of the high-fidelity model is changing, and in the latter, the low-fidelity ones.
> - **Varying the amount of low-fidelity simulations**: None of the preprints compares the performance increase/decrease with the number of low-fidelity simulations.
> - **Sequential and active learning variants**: None of the preprints has evaluations on sequential methods and active learning schemes.
> - **Code and reproducibility**: Thiele (2025) proposed to explore weight initialization with response distillation. However, to the best of our knowledge, no codebase has been released for their method. Our codebase (shared in the reproducibility statement) will be publicly available upon publication and provides guidance and a simple integration of custom simulators.
>
> We therefore believe that our work distinguishes itself from the concurrent approach in (1) the elaborate empirical evaluation, (2) discussion on the effectiveness of transfer learning in function of the similarity of the simulators, (3) code availability and usability, and (4) the contribution of sequential and active learning variants.

---

> > ### Author Response · Authors · 2025-11-21
> >
> > # Q3. The spiking network has 12/24 parameters for low/high-fidelity, curious how performance degrades as this gap increases.
> > We agree this is an important question. In response, we added an analysis in Appendix G.3 where we vary the number of LF parameters between 3, 6, and 12 (while the HF model always has 24). We find:
> >
> > Transfer learning benefits decrease as the gap between LF and HF parameter spaces increases.
> > In our setup, MF-NPE still improves over NPE for 6 LF parameters, but the gains degrade substantially once ~80% of the HF parameters are absent in the LF model.
> >
> > We now highlight this result in the main text (Sec. 4.3).
> >
> > # W1. Theoretical analyses on transfer learning bounds
> > While we agree that formal theoretical guarantees would be valuable and could further strengthen the motivation for MF-NPE, developing such guarantees would require addressing several open challenges in neural posterior estimation and transfer learning. As such, they fall outside the scope of the present work. We focus on demonstrating a flexible, scalable, and practically effective multifidelity method supported by extensive empirical validation.
> >
> > Further, we would like to offer several points of reflection on why theoretical bounds, while desirable, may not be strictly necessary for this paper:
> >
> > - **Empirical understanding of when transfer helps.** Our analysis in Sec. 4.4 (Fig. 4) provides concrete, data-driven insight into successful transfer learning. By systematically perturbing the OU process, we identify two measurable factors: mutual information and representational coherence, that strongly predict when multifidelity transfer is effective. This provides actionable guidance for practitioners, even without formal guarantees.
> >
> > - **Rigorous empirical evaluations.** For tasks with known ground-truth posteriors, we directly evaluate MF-NPE using standard metrics (C2ST, MMD). For more complex simulators, without an analytical solution, we evaluate with metrics that do not require an analytical solution (NLTP, NRMSE). We complement the experiments with calibration diagnostics (TARP, SBC) and posterior predictive checks. These evaluations offer strong empirical evidence for the validity and robustness of our approach in realistic settings.
> >
> > - **Established precedent in SBI.** Much of the progress in simulation-based inference using neural density estimators has been empirical rather than theoretical; only recently have initial theoretical results begun to emerge. Despite this, empirical advances have been accomplished and have driven progress in the field. Our work follows this tradition by providing a practical framework supported by extensive experiments across diverse domains.
> >
> > Taken together, while formal transfer-learning bounds remain an important direction for future research, we believe the empirical contributions and insights in this work already offer substantial value to the SBI community.
> >
> > # W3. Notation
> > We thank the reviewer for pointing out potential notation inconsistencies. After re-checking the manuscript, we identified a few minor issues (e.g., occasional missing boldface for vectors), which we have now corrected. Aside from these, the notation is consistent to the best of our knowledge.
> >
> > Regarding the specific example raised about switching between $p_L(x | \theta)$, $ p_H(x | \theta)$, and $p(x | \theta)$, our intention was the following. We use the subscripts L and H when referring explicitly to the low- or high-fidelity simulator, respectively, and omit the subscript and write $p(x | \theta)$ when discussing a generic simulator, or in sections where the LF and HF simulators share the same functional form and differ only through their parameter values (for example, in the controlled perturbations studied in Sec. 4.4). To enhance clarity, we now omit the H subscript, and refer to the high-fidelity simulator with the generic label $p(x | \theta)$. This allows us only to introduce the subscript L for a low-fidelity approximation.
> >
> > We thank the reviewer again for their valuable feedback, which allowed us to further improve the validation of our methods.

---

### Author Response · Authors · 2025-11-21
**General reply to reviewers**

We thank the reviewers for their thorough and insightful comments, and for acknowledging the strengths of the work. We particularly appreciate the positive feedback regarding the paper's clarity, relevance, and empirical results. Below, we directly address the key concerns raised and highlight how we have improved the manuscript in response:

# Verbosity of the introduction/background and results in appendix
Following the suggestions of reviewers G9kC and CW6v, we shortened the introduction and background section, allocating more space  to the benchmarking results and discussion on the effectiveness of transfer learning.

# Additional experimental results
We added two new experiments. First, we tested MF-NPE in the case where the  high-fidelity model has a smaller number of parameters than the low-fidelity model. We also tested, in the Spiking network model, how the performance of MF-NPE is impacted as the discrepancy in the number of parameters between the low- and high-fidelity models is systematically increased. We also provided TARP evaluations as an alternative calibration check for all the tasks.

# Clarified theoretical vs empirical contributions
While theoretical guarantees on transfer learning performance would be valuable, obtaining these results requires several substantial advances in neural posterior estimation and transfer learning. We therefore highlight this as an important direction of future work and focus the present paper on flexibility, scalability, and broad empirical validation. In lieu of theoretical guarantees, we highlight the conceptual and empirical analysis in identifying mutual information and representational coherence as key determinants of effective transfer learning.

# Naming
Following the suggestion of the reviewer G9kC, we renamed the algorithm A-MF-TSNPE to MF-TSNPE-AF, to avoid confusion with active learning or amortized inference.

We thank the reviewers again for their valuable feedback, which has greatly strengthened our manuscript. Their suggestions have helped us address important gaps, clarify key contributions, and enhance the paper's overall rigor and quality.

---

### Meta-Review · Area_Chair_2hqY · 2026-01-16

**Summary:**

This paper presents MF-(TS)NPE, a multifidelity neural posterior estimation method that uses transfer learning from cheap low-fidelity simulations to improve inference efficiency for expensive high-fidelity simulators.

The initial reviews were quite positive with three recommendations to accept (two strong) and one weak reject. Post-rebuttal scores were either explicitly stated to remain the same or inferred to remain similar.

The main strengths identified include: clear writing and motivation, practical relevance addressing a real bottleneck in SBI, comprehensive empirical evaluation across diverse tasks, and insightful analysis of when transfer learning helps (mutual information and representational coherence). The main concerns centered on: limited theoretical analysis, concurrent work reducing perceived novelty, and some presentation issues with key results initially in the appendix.

Key concerns included a lack of theoretical guarantees which remains unaddressed, and some presentation issues which were addressed in the rebuttal. There was also concern regarding concurrently published research which was also responded to a sufficient degree in the rebuttal.

**Reviewer Concerns:**

The main strengths identified include: clear writing and motivation, practical relevance addressing a real bottleneck in SBI, comprehensive empirical evaluation across diverse tasks, and insightful analysis of when transfer learning helps (mutual information and representational coherence). The main concerns centered on: limited theoretical analysis, concurrent work reducing perceived novelty, and some presentation issues with key results initially in the appendix.

Key concerns included a lack of theoretical guarantees which remains unaddressed, and some presentation issues which were addressed in the rebuttal. There was also concern regarding concurrently published research which was also responded to a sufficient degree in the rebuttal.

**Reviewer Scores:**

The initial reviews were quite positive with three recommendations to accept (two strong) and one weak reject. Post-rebuttal scores were either explicitly stated to remain the same or inferred to remain similar, with the range of scores coming from the degree to which the reviewers emphasized the strength of experimental demonstration vs the weakness of a lack of theoretical guarantees.

---

### Decision · Program_Chairs · 2026-01-26

Accept (Poster)